# TGDPO: Harnessing Token-Level Reward Guidance for Enhancing Direct Preference Optimization

**Mingkang Zhu**[1]  **Xi Chen**[2]  **Zhongdao Wang**[3]  **Bei Yu**[1]  **Hengshuang Zhao**[2]  **Jiaya Jia**[4][5]

## Abstract

Recent advancements in reinforcement learning from human feedback have shown that utilizing fine-grained token-level reward models can substantially enhance the performance of Proximal Policy Optimization (PPO) in aligning large language models. However, it is challenging to leverage such token-level reward as guidance for Direct Preference Optimization (DPO), since DPO is formulated as a sequence-level bandit problem. To address this challenge, this work decomposes the sequence-level PPO into a sequence of token-level proximal policy optimization problems and then frames the problem of token-level PPO with token-level reward guidance, from which closed-form optimal token-level policy and the corresponding token-level reward can be derived. Using the obtained reward and Bradley-Terry model, this work establishes a framework of computable loss functions with token-level reward guidance for DPO, and proposes a practical reward guidance based on the induced DPO reward. This formulation enables different tokens to exhibit varying degrees of deviation from reference policy based on their respective rewards. Experiment results demonstrate that our method achieves substantial performance improvements over DPO, with win rate gains of up to 7.5 points on MT-Bench, 6.2 points on AlpacaEval 2, and 4.3 points on Arena-Hard. Code is available at https://github.com/dvlab-research/TGDPO.

## 1. Introduction

Reinforcement Learning from Human Feedback (RLHF) has become a crucial technique for aligning Large Language models (LLMs) with human preferences and intentions (Ouyang et al., 2022; Ziegler et al., 2020). This approach has demonstrated significant success in recent LLMs advancements (OpenAI et al., 2024; Team et al., 2024a; Grattafiori et al., 2024; Team et al., 2024b). In typical RLHF workflows, a reward model is first trained using human feedback, and then the Proximal Policy Optimization (PPO) algorithm (Schulman et al., 2017) is employed to fine-tune the policy model. Typically, in these methods, a sequence-level reward is assigned to the last token of a sequence. However, this approach faces challenges, such as the sparse reward problem (i.e., delayed feedback), which leads to instability and sample inefficiency in PPO training (Choshen et al., 2020). This issue is particularly pronounced in LLM training, where responses are often lengthy and generated at the token level (Yang et al., 2023). Recent research has suggested that leveraging dense token-level reward models (Yang et al., 2023; Yin et al., 2025; Zhong et al., 2024) can help alleviate these issues, improving PPO's performance in aligning LLMs with human preferences.

Recent developments in RLHF have centered around creating simpler and more efficient algorithms that eliminate the need for a separate reward model. A notable approach in this direction is Direct Preference Optimization (DPO) (Rafailov et al., 2023). DPO reparameterizes the reward function in RLHF by directly using preference data to optimize the policy model, bypassing the traditionally required step of training a separate reward model. This reparameterization streamlines the alignment process, making DPO a popular algorithm for LLM alignment. While dense token-level reward guidance has been proved beneficial for PPO (Yang et al., 2023; Yin et al., 2025; Zhong et al., 2024), its extension to DPO is nontrivial, as DPO is formulated as a sequence-level bandit problem. In this context, the reward is expressed through the policy being optimized, and integrating token-level reward guidance into this framework presents a significant challenge, especially in eliminating the partition function from the loss function.

To fill this gap, we decompose the sequence-level proximal

---

[1]The Chinese University of Hong Kong [2]The University of Hong Kong [3]Huawei [4]SmartMore [5]The Hong Kong University of Science and Technology. Correspondence to: Mingkang Zhu <mkzhu23@cse.cuhk.edu.hk>, Jiaya Jia <jia@cse.ust.hk>.

*Proceedings of the 42$^{nd}$ International Conference on Machine Learning*, Vancouver, Canada. PMLR 267, 2025. Copyright 2025 by the author(s).

policy optimization into a sequence of token-level proximal policy optimization problems and modify them to incorporate token-level reward guidance. We derive a closed-form optimal token-level policy and the corresponding token-level reward for the modified problem. Based on the obtained reward and Bradley-Terry model, especially a new theoretical result for eliminating partition function, we propose a preference optimization algorithm framework with token-level reward guidance for DPO, which we refer to as TGDPO. Additionally, we introduce a practical token-level reward guidance based on the induced DPO reward.

Extensive experiments are conducted on three instruction following benchmarks: AlpacaEval 2 (Li et al., 2023), MT-Bench (Zheng et al., 2023), and Arena-Hard (Li et al., 2024). TGDPO consistently outperforms existing preference optimization algorithms, achieving improvements of up to 7.5 points on MT-Bench, 6.2 points on AlpacaEval 2, and 4.3 points on Arena-Hard compared to the best baseline method. We further demonstrate and analyze the unique advantages of TGDPO. We empirically show that TGDPO achieves satisfactory policies upon loss convergence, which is not commonly observed in conventional preference optimization methods. TGDPO also enables control over convergence speed and is robust to variations in token-level rewards. These properties significantly enhance the efficiency and practicality of the algorithm. Our key contributions are outlined below:

- We decompose the sequence-level PPO into a sequence of token-level proximal policy optimization problems via the upper-bounding approach and derive a closed-form optimal token-level policy for the modified problem, with which the corresponding reward can be represented along with the token-level reward guidance.

- With the obtained reward, the Bradley-Terry model, and a new result for eliminating the partition function, we propose TGDPO, a preference optimization algorithm framework with token-level reward guidance for DPO. We further introduce a practical token-level reward guidance based on the induced DPO reward.

- Extensive experiments demonstrate that our TGDPO improves win rates by up to 7.5 points on MT-Bench, 6.2 points on AlpacaEval 2, and 4.3 points on Arena-Hard compared to the best baseline.

## 2. Related Work

**Reinforcement Learning from Human Feedback.** Reinforcement learning from human feedback (RLHF) has been extensively applied for aligning LLMs with human preferences and values (Ouyang et al., 2022; Ziegler et al., 2020). The standard RLHF pipeline typically consists of two stages: reward modeling and policy optimization through reinforcement learning. Proximal Policy Optimization (PPO) with on-policy sampling (Schulman et al., 2017) is commonly used for this purpose. However, challenges in effective reward modeling and tuning the PPO algorithm to achieve optimal performance have motivated alternative approaches that bypass the reward modeling step and focus on directly optimizing the policy. The direct preference optimization (DPO) algorithm (Rafailov et al., 2023) is a representative one. DPO explicitly represents the reward function with the optimal policy of the proximal policy optimization problem, thereby avoiding the need for a separate reward model and fine-tuning LLMs directly with human preference. DPO has proven to be both lightweight and stable, showing strong performance in a range of applications (Ivison et al., 2024; Tian et al., 2024; Miao et al., 2024). Several variants of DPO have since been proposed, improving its performance. For instance, R-DPO (Park et al., 2024) addresses DPO's tendency to exploit token length, while SimPO (Meng et al., 2024) aims to better align the objective with the decoding formula and eliminate the need for a reference model. KTO (Ethayarajh et al., 2024) focuses on optimizing preferences using non-pairwise data. These preference optimization techniques, however, operate at the sequence level and do not shape the reward function of DPO from the token level. In contrast, our approach aims to leverage token-level rewards to guide preference optimization and better align LLMs. A recent work TDPO (Zeng et al., 2024) tries to provide a token-level understanding of DPO. It explains DPO using token-level Markov decision process and proposes to incorporate forward KL divergence to the DPO objective. However, like DPO, TDPO still does not consider token-level reward guidance. Our TGDPO, on the other hand, explicitly incorporates token-level reward signals into the preference optimization framework.

**RLHF with Dense Token-Level Reward.** Text generation of LLMs can be modeled as a Markov decision process. Sequence-level PPO treats the entire sequence as an action and assigns a reward at the sequence's end (Schulman et al., 2017), which results in sparse feedback at the token level. This sparsity hinders the model's ability to differentiate between preferred and dispreferred tokens within a sequence, leading to training instability (Snell et al., 2023; Xia et al., 2024). To mitigate this issue, several techniques have been developed to generate dense token-level rewards, including learning from fine-grained human feedback (Wu et al., 2023), fine-grained AI feedback (Ouyang et al., 2024), and grounding preferences at the token or segment level (Yang et al., 2023; Yin et al., 2025; Zhong et al., 2024). PPO leveraging such fine-grained reward models has shown significant performance improvements. However, extending token-level guidance to DPO is a challenge, as DPO's reward function is explicitly expressed through the policy being optimized. Incorporating token-level reward guidance

into the DPO framework requires overcoming substantial difficulties, especially in eliminating the partition function from the loss function, which remains an open problem. More discussions on closely related work are presented in Appendix C.

## 3. Preliminary

Given a human preference dataset $\mathcal{D} = \{(x, y_w, y_l)\}$, where $x$ is a prompt, $y_w$ and $y_l$ are preferred and dispreferred responses respectively, in RLHF a sequence-level reward model $r_\phi(x, y)$ is first trained with the preference dataset for assigning higher reward to preferred response and lower reward to dispreferred one. With the trained reward model, sequence-level Proximal Policy Optimization (PPO) solves the following problem to fine-tune LLMs:

$$\max_{\pi_\theta} \mathbb{E}_{x \sim \mathcal{D}, y \sim \pi_\theta(\cdot|x)} \left[ r_\phi(x, y) \right] - \beta \mathbb{D}_{\mathrm{KL}}[\pi_\theta(\cdot|x)||\pi_{\mathrm{ref}}(\cdot|x)]$$

$$= \max_{\pi_\theta} \mathbb{E}_{x \sim \mathcal{D}, y \sim \pi_\theta(\cdot|x)} \left[ r_\phi(x, y) - \beta \log \frac{\pi_\theta(y|x)}{\pi_{\mathrm{ref}}(y|x)} \right], \tag{1}$$

where $\mathbb{D}_{\mathrm{KL}}[\cdot]$ is the KL-divergence of two probability distributions, $\pi_\theta$ is the language model policy, $\pi_{\mathrm{ref}}$ is the reference policy, and the positive parameter $\beta$ controls the deviation of $\pi_\theta$ from $\pi_{\mathrm{ref}}$. Equation (1) can be considered as assigning the reward to a sequence and is referred to as the sequence-level PPO problem in this work. It has the issue of sparse reward (delayed feedback) that challenges traditional deep reinforcement learning (Andrychowicz et al., 2017). To alleviate the issue, sequence-level PPO with token-level reward guidance is developed to fine-tune LLMs in a fine-grained fashion with dense token-wise rewards (Yang et al., 2023; Yin et al., 2025; Zhong et al., 2024).

**Sequence-Level PPO with Token-Level Reward Guidance.** Text generation of an LLM can be modeled as a Markov Decision Process (MDP). Let $s_t$ be the context for generating the token at time step $t \geq 0$, the generated token is denoted as $a_t \sim \pi_\theta(\cdot|s_t)$. For a prompt $x$ of the LLM, $s_0 = x$ and $s_t = [x, a^{<t}]$, where $a^{<t} = [a_0, \ldots, a_{t-1}]$ are the previously generated tokens. The generated full text-sequence with $T$ tokens is denoted as $\boldsymbol{a} = [a_0, \ldots, a_{T-1}]$. A token-level reward, for convenience it is also denoted by $r_\phi(s_t, a_t)$, is learned so that the reward sequence is dense and can guide the selection of token at any time step, which is called token-level reward guidance (Yang et al., 2023; Yin et al., 2025). Typically, the problem of sequence-level proximal policy optimization with token-level reward guidance is (Yin et al., 2025):

$$\max_{\pi_\theta} \mathbb{E}_{x \sim \mathcal{D}, y \sim \prod_{t=0}^{T-1} \pi_\theta(a_t|s_t)} \left[ \sum_{t=0}^{T-1} r_\phi(s_t, a_t) - \beta \log \frac{\pi_\theta(y|x)}{\pi_{\mathrm{ref}}(y|x)} \right], \tag{2}$$

where $x$ is a prompt, $s_t$ and $a_t$ are the state and action defined previously, $y = [a_0, \ldots, a_{T-1}]$ is the response generated by $\pi_\theta$ from the given prompt $x$. Classically, the sequence-level reward function $r_\phi(x, y)$ can be set as $r_\phi(x, y) = \sum_{t=0}^{T-1} r_\phi(s_t, a_t)$ (Yang et al., 2023).

**Direct Preference Optimization.** Direct preference optimization (Rafailov et al., 2023) bypasses learning a reward model and aligns directly an LLM to human preference. DPO (Rafailov et al., 2023) expresses the sequence-level reward function explicitly with the optimal policy of Equation (1) as:

$$r_\phi(x, y) = \beta \log \frac{\pi_\theta(y|x)}{\pi_{\mathrm{ref}}(y|x)} + \beta \log Z(x), \tag{3}$$

where $Z(x)$ is the partition function and $\beta$ is a positive constant. By adopting the Bradley-Terry preference model (Bradley & Terry, 1952)

$$\Pr(y_w \succ y_l|x) = \frac{\exp\left(r_\phi(x, y_w)\right)}{\exp\left(r_\phi(x, y_w)\right) + \exp\left(r_\phi(x, y_l)\right)} \tag{4}$$

for specifying human preference distribution, DPO obtains the following loss function:

$$\mathcal{L}_{\mathrm{DPO}}(\pi_\theta) = -\mathbb{E}_{(x, y_w, y_l) \sim \mathcal{D}} \left[ \log \sigma \left( \beta \log \frac{\pi_\theta(y_w|x)}{\pi_{\mathrm{ref}}(y_w|x)} \right. \right.$$
$$\left. \left. - \beta \log \frac{\pi_\theta(y_l|x)}{\pi_{\mathrm{ref}}(y_l|x)} \right) \right], \tag{5}$$

which is obtained by substituting Equation (3) into Equation (4), where $\sigma$ is the sigmoid function. DPO minimizes Equation (5) with respect to the policy $\pi_\theta$ to directly fine-tune the LLM with the preference dataset at the sequence level.

## 4. Methodology

Direct preference optimization expresses the reward function explicitly with the optimal policy of the sequence-level proximal policy optimization problem. However, incorporating existing token-level rewards explicitly into DPO to guide fine-tuning is an unresolved problem. To derive a form of DPO with token-level reward guidance, this section first gives the problem of token-level PPO in Section 4.1 from the sequence-level PPO in Equation (2). The token-level PPO problem is further modified to incorporate token-level reward guidance in Section 4.2, the closed-form optimal policy is derived, and the corresponding token-level reward with guidance is obtained. Then with the Bradley-Terry model, we propose the direct preference optimization with token-level reward guidance in Section 4.3.

### 4.1. Token-Level PPO

Note that $y = [a_0, \ldots, a_{T-1}]$ is the response generated by $\pi_\theta$ from the given prompt $x$. Using the notations of state

and action in Section 3, we can get

$$\pi_\theta(y|x) = \pi_\theta([a_0, \ldots, a_{T-1}]|x) = \prod_{t=0}^{T-1} \pi_\theta(a_t|s_t);$$

$$\pi_{\text{ref}}(y|x) = \pi_{\text{ref}}([a_0, \ldots, a_{T-1}]|x) = \prod_{t=0}^{T-1} \pi_{\text{ref}}(a_t|s_t).$$

Thus, the objective function in Equation (2) can be decomposed into the token level as:

$$\sum_{t=0}^{T-1} r_\phi(s_t, a_t) - \beta \log \frac{\pi_\theta(y|x)}{\pi_{\text{ref}}(y|x)}$$
$$= \sum_{t=0}^{T-1} \left( r_\phi(s_t, a_t) - \beta \log \frac{\pi_\theta(a_t|s_t)}{\pi_{\text{ref}}(a_t|s_t)} \right). \quad (6)$$

Moreover, according to the MDP for language model (Section 3), $y \sim \prod_{t=0}^{T-1} \pi_\theta(a_t|s_t)$ in Equation (2) is equivalent to $y \sim \pi_\theta(\cdot|x)$, which is further equivalent to $s_0 = x \sim \mathcal{D}$, $a_t \sim \pi_\theta(\cdot|s_t)$, $t = 0, 1, \ldots, T-1$. Then by Equation (6), the problem of sequence-level PPO with token-level reward guidance in Equation (2) becomes

$$\max_{\pi_\theta} \mathbb{E}_{x \sim \mathcal{D}, y \sim \pi_\theta(\cdot|x)} \left[ \sum_{t=0}^{T-1} (r_\phi(s_t, a_t) - \beta \log \frac{\pi_\theta(a_t|s_t)}{\pi_{\text{ref}}(a_t|s_t)}) \right]$$
$$= \max_{\pi_\theta} \mathbb{E}_{s_0 \sim \mathcal{D}, a_t \sim \pi_\theta(\cdot|s_t), t=0,1,\ldots,T-1} \left[ \sum_{t=0}^{T-1} (r_\phi(s_t, a_t) \right.$$
$$\left. - \beta \log \frac{\pi_\theta(a_t|s_t)}{\pi_{\text{ref}}(a_t|s_t)}) \right]. \quad (7)$$

Based on Equation (7), we can show that:

**Theorem 4.1.** *The maximum value of the sequence-level proximal policy optimization in Equation (2) is upper bounded by the summation from $t = 0, 1, \ldots, T-1$ of the maximum value of the problem:*

$$\max_{\pi_\theta} \mathbb{E}_{s_t \sim \mathcal{D}_t, a_t \sim \pi_\theta(\cdot|s_t)} \left[ r_\phi(s_t, a_t) - \beta \log \frac{\pi_\theta(a_t|s_t)}{\pi_{ref}(a_t|s_t)} \right] \quad (8)$$

*where $s_t \sim \mathcal{D}_t$ denotes that $s_0 = x \sim \mathcal{D}$ and $a_p \sim \pi_\theta(\cdot|s_p)$, $p = 0, 1, \ldots, t-1$.*

The proof of Theorem 4.1 is given in Appendix A.1.

Equation (8) is the problem of token-level PPO at time step $t$, which optimizes the policy for action $a_t$ given the state $s_t$. Theorem 4.1 suggests that, the sequence-level proximal policy optimization in Equation (2) can be upper-bounded with a sequence of token-level PPOs in Equation (8). However, it

is not easy to solve the problem since $s_t \sim \mathcal{D}_t$ is dependent on the policy $\pi_\theta$ to be optimized (see Equation (1) for a comparison, where the distribution $\mathcal{D}$ is independent of the policy $\pi_\theta$ to be optimized).

## 4.2. Modified Token-Level PPO with Reward Guidance and Optimal Policy

Given win and lose responses $y_w = (a_0^w, \ldots, a_{T_w-1}^w)$ and $y_l = (a_0^l, \ldots, a_{T_l-1}^l)$, Rafailov et al. (2024) expressed the per-instance loss of DPO (Rafailov et al., 2023) in the token-level as:

$$\Pr(y_w \succ y_l)$$
$$= \sigma \left( \sum_{t=0}^{T_w-1} \beta \log \frac{\pi_\theta(a_t^w|s_t^w)}{\pi_{\text{ref}}(a_t^w|s_t^w)} - \sum_{t=0}^{T_l-1} \beta \log \frac{\pi_\theta(a_t^l|s_t^l)}{\pi_{\text{ref}}(a_t^l|s_t^l)} \right).$$

Assuming access to a token-level reward $\hat{r}(s_t, a_t)$, since the token-level reward $\hat{r}(s_t, a_t)$ may imply whether the action $a_t$ is preferred or dispreferred in the state $s_t$, this work aims to replace $\beta$ in the above equation with $\beta f(\hat{r}(s_t, a_t))$, a function of the token-level reward $\hat{r}(s_t, a_t)$, to guide the DPO.

Following DPO (Rafailov et al., 2023), we derive this form of loss function from the token-level proximal policy optimization in Equation (8) by incorporating the token-level reward guidance $f(\hat{r}(s_t, a_t))$. First, similar to (Zeng et al., 2024; Yang et al., 2024), we relax $s_t \sim \mathcal{D}_t$ to $s_t \sim \mathcal{D}$ and make Equation (8) solvable as

$$\max_{\pi_\theta} \mathbb{E}_{s_t \sim \mathcal{D}, a_t \sim \pi_\theta(\cdot|s_t)} \left[ r_\phi(s_t, a_t) - \beta \log \frac{\pi_\theta(a_t|s_t)}{\pi_{\text{ref}}(a_t|s_t)} \right]. \quad (9)$$

Next, we manage to incorporate token-level reward guidance $f(\hat{r}(s_t, a_t))$ into this formulation, and represent the ground-truth unknown reward function $r_\phi(s_t, a_t)$ with the optimal policy of this equation. The obtained ground-truth reward $r_\phi(s_t, a_t)$ is subsequently leveraged to construct our DPO's loss function under the Bradley-Terry preference model.

Directly replacing $\beta$ in Equation (9) with $\beta f(\hat{r}(s_t, a_t))$ might not make the problem easy to solve. To address this issue, by noting that $\beta$ is a positive constant, Equation (9) is equivalent to

$$\max_{\pi_\theta} \mathbb{E}_{s_t \sim \mathcal{D}, a_t \sim \pi_\theta(\cdot|s_t)} \left[ \frac{r_\phi(s_t, a_t)}{\beta} - \log \frac{\pi_\theta(a_t|s_t)}{\pi_{\text{ref}}(a_t|s_t)} \right]. \quad (10)$$

Then, we make the following Assumption 4.2 for incorporating token-level reward guidance $f(\hat{r}(s_t, a_t))$ explicitly into Equation (10).

**Assumption 4.2.** Suppose we have an existing reward model $\hat{r}(\cdot)$, which can generate a dense token-level reward

sequence $\hat{r}(s_t, a_t)$, $t = 0, 1, \ldots, T - 1$. Moreover, suppose $f(u)$ is a positive univariate function of $u$.

It was shown in Rafailov et al. (2024) under the definition of equivalent state-action reward class and invariant reparameterization that, DPO implicitly learns a token-level reward $\hat{r}(s_t, a_t)$ of the form $\beta \log \frac{\pi_\theta(a_t|s_t)}{\pi_{\text{ref}}(a_t|s_t)}$, and the total reward $\hat{r}(x, y) = \sum_{t=0}^{T-1} \hat{r}(s_t, a_t)$. Hence Assumption 4.2 is feasible.

**Modified Token-Level PPO.** With Assumption 4.2, we propose to adopt the token-level reward $\hat{r}(s_t, a_t)$ to guide token-level PPO. First, the parameter $\beta$ in Equation (10) is replaced with $\beta f(\hat{r}(s_t, a_t))$ and we obtain the modified problem of token-level PPO with token-level reward guidance as follows:

$$\max_{\pi_\theta} \mathbb{E}_{s_t \sim \mathcal{D}, a_t \sim \pi_\theta(\cdot|s_t)} \left[ \frac{r_\phi(s_t, a_t)}{\beta f(\hat{r}(s_t, a_t))} - \log \frac{\pi_\theta(a_t|s_t)}{\pi_{\text{ref}}(a_t|s_t)} \right],$$

$$(11)$$

where $f(\hat{r}(s_t, a_t))$ with the token-level reward $\hat{r}(s_t, a_t)$ is adopted to modify the ground-truth unknown reward function $r_\phi(s_t, a_t)$.

Thus similar to (Rafailov et al., 2023), the optimal policy for the action $a_t$ at time step $t$ of the modified token-level proximal policy optimization in Equation (11) can be derived as the following Theorem 4.3.

**Theorem 4.3.** *The optimal policy $\pi_{\theta_t}(a_t|s_t)$ for the action $a_t$ at time step $t$ of the modified token-level proximal policy optimization in Equation (11) is*

$$\pi_{\theta_t}(a_t|s_t) = \frac{\pi_{ref}(a_t|s_t) \exp\left(\frac{r_\phi(s_t, a_t)}{\beta f(\hat{r}(s_t, a_t))}\right)}{Z(s_t)},$$

*where $Z(s_t) = \mathbb{E}_{a_t \sim \pi_{ref}(\cdot|s_t)} \left[ \exp\left(\frac{r_\phi(s_t, a_t)}{\beta f(\hat{r}(s_t, a_t))}\right) \right]$ is the partition function, and $s_t \sim \mathcal{D}$ does not depend on $\pi_{\theta_t}$. Moreover, the ground-truth unknown token-level reward can be represented with the optimal policy $\pi_{\theta_t}(a_t|s_t)$ as:*

$$\frac{r_\phi(s_t, a_t)}{f(\hat{r}(s_t, a_t))} = \beta \log \frac{\pi_{\theta_t}(a_t|s_t)}{\pi_{ref}(a_t|s_t)} + \beta \log Z(s_t). \quad (12)$$

The proof of Theorem 4.3 is provided in Appendix A.2.

**Modified Token-Level Reward.** By Equation (12), we have the token-level reward function

$$r_\phi(s_t, a_t) = \beta f(\hat{r}(s_t, a_t)) \log \frac{\pi_{\theta_t}(a_t|s_t)}{\pi_{\text{ref}}(a_t|s_t)} + \beta f(\hat{r}(s_t, a_t)) \log Z(s_t), \quad (13)$$

where $f(\hat{r}(s_t, a_t))$ satisfies Assumption 4.2, $\beta$ is a constant, $s_t \sim \mathcal{D}$ does not depend on $\pi_{\theta_t}$, $t = 0, 1, \ldots, T - 1$.

Without loss of generality, suppose that trajectories generated by LLMs are bounded by a finite number of time steps, or tokens. Then, since LLMs are over-parameterized, we may assume without loss of generality that, there exists $\theta$ such that $\pi_\theta(a_t|s_t) = \pi_{\theta_t}(a_t|s_t)$, $t = 0, 1, \ldots, T - 1$. Thus, with the notations of the prompt $x$ and the generated sequence $y$, Equation (13) can be rewritten in the form

$$r_\phi([x, y^{<t}], y^t) = \beta f(\hat{r}([x, y^{<t}], y^t)) \log \frac{\pi_\theta(y^t|[x, y^{<t}])}{\pi_{\text{ref}}(y^t|[x, y^{<t}])} + \beta f(\hat{r}([x, y^{<t}], y^t)) \log Z([x, y^{<t}])$$

$$(14)$$

for all time-step $t$, where the last term with the partition function does not depend on $\pi_\theta$, according to Theorem 4.3.

### 4.3. Direct Preference Optimization with Token-Level Reward Guidance

For the proximal policy optimization with token-level reward guidance in Equation (11), Section 4.2 has represented the ground-truth unknown token-level reward $r_\phi(s_t, a_t)$ explicitly in Equation (14). Subsequently, the total reward $r_\phi(x, y)$ for the prompt $x$ and its response $y$ can be expressed as:

$$r_\phi(x, y) = \sum_{t=0}^{T} \beta f(\hat{r}([x, y^{<t}], y^t)) \log \frac{\pi_\theta(y^t|[x, y^{<t}])}{\pi_{\text{ref}}(y^t|[x, y^{<t}])} + \sum_{t=0}^{T} \beta f(\hat{r}([x, y^{<t}], y^t)) \log Z([x, y^{<t}]),$$

$$(15)$$

where the last term with the partition function does not depend on $\pi_\theta$.

Next, we derive the loss function with token-level reward guidance for direct preference optimization, as we set the target at the beginning of Section 4.2. Given a human preference dataset $\mathcal{D} = \{(x, y_w, y_l)\}$, where $x$ is a prompt, $y_w$ and $y_l$ are preferred and dispreferred responses respectively, we adopt the reward function in Equation (15) and the Bradley-Terry preference model in Equation (4) for specifying human preference. To this aim, we choose different shaping functions $f_w(\cdot)$ and $f_l(\cdot)$ for win and lose responses respectively, both of them satisfy the condition in Assumption 4.2. Then by substituting Equation (15) into Equation (4), we can get the per-instance loss detailed as follows.

**Bradley-Terry Model with Token-Level Reward Guidance.** From Equation (15), for convenience we let

$$\varphi(\pi_\theta, f, \hat{r}; x, y_w, y_l)$$

$$= \sum_{t=0}^{T_w - 1} \beta f_w(\hat{r}([x, y_w^{<t}], y_w^t)) \log \frac{\pi_\theta(y_w^t|[x, y_w^{<t}])}{\pi_{\text{ref}}(y_w^t|[x, y_w^{<t}])}$$

$$- \sum_{t=0}^{T_l-1} \beta f_l(\hat{r}([x, y_l^{<t}], y_l^t)) \log \frac{\pi_\theta(y_l^t|[x, y_l^{<t}])}{\pi_{\text{ref}}(y_l^t|[x, y_l^{<t}])}; \quad (16)$$

$$\delta(f, \hat{r}; x, y_w, y_l)$$
$$= \sum_{t=0}^{T_w-1} \beta f_w(\hat{r}([x, y_w^{<t}], y_w^t)) \log Z([x, y_w^{<t}])$$
$$- \sum_{t=0}^{T_l-1} \beta f_l(\hat{r}([x, y_l^{<t}], y_l^t)) \log Z([x, y_l^{<t}]),$$

where $T_w$ and $T_l$ are the lengths of the responses $y_w$ and $y_l$ respectively. Then, the Bradley-Terry preference model with token-level reward guidance is

$$\Pr(y_w \succ y_l|x)$$
$$= \sigma\left(\varphi(\pi_\theta, f, \hat{r}; x, y_w, y_l) + \delta(f, \hat{r}; x, y_w, y_l)\right). \quad (17)$$

The proof of Equation (17) is given in Appendix A.3.

The above function is not computable since it contains partition functions in $\delta(f, \hat{r}; x, y_w, y_l)$. Notably, preference optimization aims to maximize the preference function in Equation (17) with respect to $\pi_\theta$, and $\delta(f, \hat{r}; x, y_w, y_l)$ does not depend on the policy $\pi_\theta$, we can eliminate $\delta(f, \hat{r}; x, y_w, y_l)$ from Equation (17) based on the following Theorem 4.4.

**Theorem 4.4.** *The preference function in Equation* (17) *has the same maxima and the same ascent directions as the function* $\sigma\left(\varphi(\pi_\theta, f, \hat{r}; x, y_w, y_l)\right)$. *Moreover, for two policies* $\pi_{\theta_1}$ *and* $\pi_{\theta_2}$,

$$\sigma\left(\varphi(\pi_{\theta_1}, f, \hat{r}; x, y_w, y_l) + \delta(f, \hat{r}; x, y_w, y_l)\right)$$
$$> \sigma\left(\varphi(\pi_{\theta_2}, f, \hat{r}; x, y_w, y_l) + \delta(f, \hat{r}; x, y_w, y_l)\right) \quad (18)$$

*if and only if*

$$\sigma\left(\varphi(\pi_{\theta_1}, f, \hat{r}; x, y_w, y_l)\right)$$
$$> \sigma\left(\varphi(\pi_{\theta_2}, f, \hat{r}; x, y_w, y_l)\right). \quad (19)$$

The proof of Theorem 4.4 is given in Appendix A.4. Theorem 4.4 is due to that, the sigmoid function is strictly increasing and it does not change the order of values. Hence Theorem 4.4 suggests that, maximizing $\sigma\left(\varphi(\pi_\theta, f, \hat{r}; x, y_w, y_l)\right)$ with respect to $\pi_\theta$ is equivalent to maximizing the preference function in Equation (17) with respect to $\pi_\theta$. Furthermore, the equivalence between Equation (18) and Equation (19) demonstrates that, for any two policies $\pi_{\theta_1}$ and $\pi_{\theta_2}$, canceling the term $\delta(f, \hat{r}; x, y_w, y_l)$ from Equation (18) does not affect the preference order of the responses $y_w$ and $y_l$.

**Loss Function.** Since we only care about the optimal policy of Equation (17), by Theorem 4.4 we may redefine the

preference function as $\sigma\left(\varphi(\pi_\theta, f, \hat{r}; x, y_w, y_l)\right)$, i.e.,

$$\Pr(y_w \succ y_l|x) \triangleq \sigma\left(\varphi(\pi_\theta, f, \hat{r}; x, y_w, y_l)\right)$$
$$= \sigma\left(\sum_{t=0}^{T_w-1} \beta f_w(\hat{r}([x, y_w^{<t}], y_w^t)) \log \frac{\pi_\theta(y_w^t|[x, y_w^{<t}])}{\pi_{\text{ref}}(y_w^t|[x, y_w^{<t}])}\right.$$
$$\left. - \sum_{t=0}^{T_l-1} \beta f_l(\hat{r}([x, y_l^{<t}], y_l^t)) \log \frac{\pi_\theta(y_l^t|[x, y_l^{<t}])}{\pi_{\text{ref}}(y_l^t|[x, y_l^{<t}])}\right),$$

which specifies the per-instance human preference and is computable. Furthermore, analogous to Equation (5), we formulate the loss function for enhancing DPO by harnessing token-level reward guidance as follows:

$$\mathcal{L}_{\text{TGDPO}}(\pi_\theta) = -\mathbb{E}_{(x, y_w, y_l) \sim \mathcal{D}}\left[\log \sigma\left(\sum_{t=0}^{T_w-1}\right.\right.$$
$$\beta \cdot f_w(\hat{r}([x, y_w^{<t}], y_w^t)) \cdot \log \frac{\pi_\theta(y_w^t|[x, y_w^{<t}])}{\pi_{\text{ref}}(y_w^t|[x, y_w^{<t}])} -$$
$$\left.\left. \sum_{t=0}^{T_l-1} \beta f_l(\hat{r}([x, y_l^{<t}], y_l^t)) \log \frac{\pi_\theta(y_l^t|[x, y_l^{<t}])}{\pi_{\text{ref}}(y_l^t|[x, y_l^{<t}])}\right)\right]. \quad (20)$$

The loss function $\mathcal{L}_{\text{TGDPO}}(\pi_\theta)$ in Equation (20) provides a framework of direct preference optimization, by leveraging $f(\hat{r}(s_t, a_t))$ to shape the optimization of the policy on the tokens of win and lose responses. Specifically, with an appropriate choice of $f(\cdot)$, this framework can recover several known direct preference optimization methods. For example, if we take $f_w \equiv f_l \equiv 1$, then Equation (20) is the loss function of DPO (Rafailov et al., 2023) (for others, see Appendix C.2). Nonetheless, the aim of this framework is to use token-level reward $\hat{r}(s_t, a_t)$ to shape the loss function in Equation (20) directly. In the following, we provide a practical example.

**Practical Method.** For convenience, we adopt the induced DPO reward (Rafailov et al., 2023) for the token-level reward $\hat{r}(s_t, a_t)$. Suppose $\pi_{\hat{\theta}}$ is an optimal policy of the loss function of DPO in Equation (5), Rafailov et al. (2024) showed in Theorem 1 that DPO learns implicitly a token-level reward of the form

$$\hat{r}([x, y^{<t}], y^t) = \beta \log \frac{\pi_{\hat{\theta}}(y^t|[x, y^{<t}])}{\pi_{\text{ref}}(y^t|[x, y^{<t}])}.$$

Hence for Equation (20), we simply set

$$f_w(\hat{r}([x, y_w^{<t}], y_w^t)) = 1 + \alpha \, \hat{r}([x, y_w^{<t}], y_w^t);$$
$$f_l(\hat{r}([x, y_l^{<t}], y_l^t)) = 1 - \alpha \, \hat{r}([x, y_l^{<t}], y_l^t), \quad (21)$$

where $\alpha$ is a positive constant. Obviously, this setting meets Assumption 4.2 if $\alpha$ is small enough.

**Motivation of the Practical Method.** Observing the loss function $\mathcal{L}_{\text{TGDPO}}(\pi_\theta)$ in Equation (20), below is the motivation for setting $f(\hat{r}([x, y^{<t}], y^t))$ as in Equation (21):

- For a token $y_w^t$ in win-response, if $\hat{r}([x, y_w^{<t}], y_w^t) > 0$, then it is identified as a preferred token, implying that the state-action should be reinforced, and then it is assigned a larger weight $1 + \alpha \hat{r}([x, y_w^{<t}], y_w^t)$. In this way, the gradient of our loss function $\mathcal{L}_{\text{TGDPO}}(\pi_\theta)$ at this state-action is

$$\beta(1 + \alpha \hat{r}([x, y_w^{<t}], y_w^t)) \nabla_{\pi_\theta} \log \frac{\pi_\theta(y_w^t | [x, y_w^{<t}])}{\pi_{\text{ref}}(y_w^t | [x, y_w^{<t}])},$$

which is scaled up by $1 + \alpha \hat{r}([x, y_w^{<t}], y_w^t)$. As a result, optimizing our loss function $\mathcal{L}_{\text{TGDPO}}(\pi_\theta)$ encourages the policy to assign a higher probability to this action.

- Similarly, the token $y_w^t$ satisfying $\hat{r}([x, y_w^{<t}], y_w^t) < 0$ is identified as a dispreferred token, although it is in the preferred response $y_w$. Then by assigning weight $1 + \alpha \hat{r}([x, y_w^{<t}], y_w^t) < 1$, optimizing our loss function $\mathcal{L}_{\text{TGDPO}}(\pi_\theta)$ would progressively assign a lower probability to this action.

- For a token $y_l^t$ in lose-response, if $\hat{r}([x, y_l^{<t}], y_l^t) < 0$, then it is considered as a dispreferred token. Thus since the weight $1 - \alpha \hat{r}([x, y_l^{<t}], y_l^t)) > 1$, optimizing the loss function $\mathcal{L}_{\text{TGDPO}}(\pi_\theta)$ would assign an even lower probability to this action.

- The token $y_l^t$ satisfying $\hat{r}([x, y_l^{<t}], y_l^t) > 0$ is considered as a preferred token, although it is in the dispreferred response $y_l$. In this case $1 - \alpha \hat{r}([x, y_l^{<t}], y_l^t)) < 1$, then optimizing the loss function $\mathcal{L}_{\text{TGDPO}}(\pi_\theta)$ would progressively assign a higher probability to this action.

The above analysis indicates that our direct preference optimization with token-level reward guidance performs in the token-level granularity, and exhibits varying degrees of deviation from the reference policy based on their respective rewards. This property inherently empowers our approach to discover satisfactory policies, leading to better policies than existing approaches. This property should be attributed to the modified token-level PPO with reward guidance in Section 4.2, and the derived loss function $\mathcal{L}_{\text{TGDPO}}(\pi_\theta)$ for direct preference optimization in Equation (20) with the setting of $f(\hat{r}([x, y^{<t}], y^t))$ in Equation (21).

## 5. Experiments

In this section, we first outline our experiment settings in Section 5.1. Then we show the main experiment results in Section 5.2. Lastly, we provide an empirical analysis of the unique properties of our TGDPO in Section 5.3.

### 5.1. Experiment Settings

**Models and Training Settings.** We conduct experiments on three models: Llama3-8B-Instruct (Grattafiori et al.,

2024), Llama3.2-3B-Instruct, and Gemma2-2B-it (Team et al., 2024b). Following (Meng et al., 2024), we use prompts from the UltraFeedback dataset (Cui et al., 2024) and let each model generate 5 responses with a temperature of 0.8. These responses are then ranked using the ArmoRM model (Wang et al., 2024). The highest and lowest-ranked responses are selected as the chosen and rejected samples, respectively. For Llama3-8B-Instruct, we further utilize the PairRM model (Jiang et al., 2023) to annotate response scores, thereby evaluating the robustness of algorithms in handling varying quality of sample annotations. Hyperparameter settings are presented in Appendix D.1.

**Evaluation Benchmarks.** We primarily evaluate trained models' performance using three widely recognized open-ended instruction-following benchmarks: MT-Bench (Zheng et al., 2023), Arena-Hard (Li et al., 2024), and AlpacaEval 2 (Li et al., 2023), which assess models' response quality across diverse queries. For MT-Bench, we report the MT-Bench score and win rate against GPT-4. For Arena-Hard, we report the win rate against GPT-4-0314. For AlpacaEval 2, we report the win rate against GPT-4 Turbo. Further details are discussed in Appendix D.2.

**Baseline Methods.** We compare our TGDPO with two state-of-the-art preference optimization methods: DPO (Rafailov et al., 2023) and SimPO (Meng et al., 2024). We also include the pre-trained Instruct model as a baseline.

### 5.2. Main Results

The experiment results on AlpacaEval 2 (Li et al., 2023), Arena-Hard (Li et al., 2024), and MT-Bench (Zheng et al., 2023) are summarized in Table 1. Our TGDPO consistently outperforms baseline methods across these benchmarks. Notably, on AlpacaEval 2, it achieves a win rate increase of up to 6.2 over the best baseline, while on MT-Bench, the win rate improves by up to 7.5. For the challenging Arena-Hard benchmark, our method demonstrates stable superior performance, with a win rate enhancement of up to 4.3 compared to the best baseline. These consistent performance improvements underscore the effectiveness of our approach. More experiment results and comparisons are presented in Appendix B.

### 5.3. Analysis

In this section, we present an empirical analysis of the unique properties of our TGDPO in comparison to conventional preference optimization approaches. The analysis is conducted under the Llama3-8B-Instruct PairRM setting.

**TGDPO Leads to Satisfactory Results upon Loss Convergence.** A well-known challenge in preference optimization algorithms is the misalignment between loss minimization and model performance (Guo et al., 2024). Specifically, min-

*Table 1.* Experiment results on AlpacaEval 2 (Li et al., 2023), Arena-Hard (Li et al., 2024), and MT-Bench (Zheng et al., 2023) benchmarks.

| Method | Llama3-8B-Instruct PairRM | | | | Llama3-8B-Instruct ArmoRM | | | |
|---|---|---|---|---|---|---|---|---|
| | AlpacaEval 2 | Arena-Hard | MT-Bench | | AlpacaEval 2 | Arena-Hard | MT-Bench | |
| | Win Rate (%) | Win Rate (%) | Score | Win Rate(%) | Win Rate (%) | Win Rate (%) | Score | Win Rate(%) |
| SFT | 30.6 | 21.4 | 7.9 | 27.5 | 30.6 | 21.4 | 7.9 | 27.5 |
| DPO | 41.7 | 30.4 | **8.0** | 37.5 | 40.8 | 36.2 | **8.2** | **46.3** |
| SimPO | 39.8 | 28.7 | 7.8 | 32.5 | 37.0 | 28.1 | 7.8 | 42.5 |
| TGDPO | **43.9** | **34.3** | **8.0** | **41.9** | **42.5** | **40.5** | 7.9 | 45.0 |

| Method | Llama3.2-3B-Instruct ArmoRM | | | | Gemma2-2B-it ArmoRM | | | |
|---|---|---|---|---|---|---|---|---|
| | AlpacaEval 2 | Arena-Hard | MT-Bench | | AlpacaEval 2 | Arena-Hard | MT-Bench | |
| | Win Rate (%) | Win Rate (%) | Score | Win Rate (%) | Win Rate (%) | Win Rate (%) | Score | Win Rate (%) |
| SFT | 23.8 | 17.1 | 7.0 | 16.3 | 32.8 | 20.1 | 7.9 | 37.5 |
| DPO | 29.6 | 23.2 | 7.9 | 29.4 | 40.8 | 26.4 | 8.0 | 43.1 |
| SimPO | 26.2 | 22.6 | 7.4 | 15.7 | 34.8 | 21.1 | 7.8 | 40.0 |
| TGDPO | **35.8** | **25.4** | **8.1** | **36.9** | **43.0** | **30.7** | **8.1** | **46.9** |

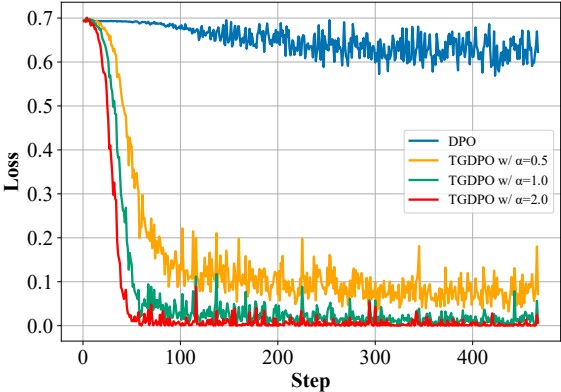

*Figure 1.* Training loss curve for DPO and our TGDPO with different values of $\alpha$. Changing the value of $\alpha$ leads to different convergence speeds for our method.

*Table 2.* Analysis of preference optimization methods' performance upon training loss convergence.

| Method | AlpacaEval 2 | Arena-Hard |
|---|---|---|
| | Win Rate (%) | Win Rate (%) |
| SFT | 30.6 | 21.4 |
| DPO | 41.7 | 30.4 |
| SimPO | 39.8 | 28.7 |
| DPO w/ convergence | 30.7 | 17.9 |
| SimPO w/ convergence | 4.6 | 2.4 |
| TGDPO w/ convergence | **43.9** | **34.3** |

*Table 3.* Analysis of our TGDPO's performance upon training loss convergence with different convergence speeds.

| Method | AlpacaEval 2 | Arena-Hard |
|---|---|---|
| | Win Rate (%) | Win Rate (%) |
| SFT | 30.6 | 21.4 |
| TGDPO w/ $\alpha = 0.5$ | **43.9** | **34.3** |
| TGDPO w/ $\alpha = 1.0$ | 42.5 | 33.9 |
| TGDPO w/ $\alpha = 2.0$ | 43.3 | **34.3** |

imizing the loss for many preference optimization methods often results in degenerate policies. This issue necessitates extensive hyperparameter tuning to identify a sweet spot between the initialization and convergence points, significantly limiting the practicality and efficiency of these algorithms. As shown in Figure 1, the optimal hyperparameters for DPO barely reduce its loss. In contrast, we empirically find that TGDPO enables convergence in much fewer steps than conventional preference optimization algorithms. In Figure 1, TGDPO demonstrates consistent and stable loss reduction toward convergence. We assume it is because TGDPO's token-level reward inherently distinguishes preferred and dispreferred tokens.

Furthermore, in Table 2, we compare benchmark performances by training each method using their default configurations and training them until loss convergence. The

results reveal that both DPO and SimPO suffer substantial performance degradation upon convergence, with SimPO's win rates dropping to single digits. Conversely, TGDPO maintains exceptional performance at the convergence point. These findings highlight the necessity of extensive hyperparameter searches for traditional preference optimization algorithms, whereas TGDPO simplifies the process, significantly improving efficiency and usability.

**TGDPO Enables Control Over Convergence Speed.** TGDPO offers the flexibility to control the speed of convergence by adjusting the value of $\alpha$ in Equation (20). A

*Table 4.* Analysis of our TGDPO's robustness using different token-level rewards $\hat{r}(s_t, a_t)$.

| Method | AlpacaEval 2 | Arena-Hard |
|---|---|---|
| | Win Rate (%) | Win Rate (%) |
| SFT | 30.6 | 21.4 |
| DPO w/ $\beta = 0.1$ | 34.8 | 26.7 |
| DPO w/ $\beta = 0.01$ | 41.7 | 30.4 |
| TGDPO w/ $\beta = 0.1$ for $\hat{r}(s_t, a_t)$ | 42.8 | **34.3** |
| TGDPO w/ $\beta = 0.01$ for $\hat{r}(s_t, a_t)$ | **43.9** | **34.3** |

larger $\alpha$ provides stronger token-level guidance, resulting in faster convergence, while a smaller $\alpha$ aligns the algorithm more closely with conventional DPO behavior. As illustrated in Figure 1, increasing $\alpha$ leads to a more rapid loss reduction compared to lower values of $\alpha$. Additionally, in Table 3, we compare benchmark performances at the respective convergence points for different values of $\alpha$. Specifically, we evaluate checkpoints at step 50 for $\alpha = 2.0$, step 60 for $\alpha = 1.0$, and epoch 1 for $\alpha = 0.5$. The results demonstrate comparable performance across all configurations, especially for the challenging Arena-Hard benchmark. This desirable property of TGDPO allows for early stopping once the loss converges, significantly reducing computational costs without compromising performance.

**TGDPO is Robust to Variations in Token-Level Rewards $\hat{r}(s_t, a_t)$.** To make TGDPO practical, we propose using token-level rewards derived from pre-trained DPO models as a convenient implementation. A key question arises: how sensitive is TGDPO to the quality of the token-level rewards $\hat{r}(s_t, a_t)$ defined in Equation (20)? To investigate this, we analyze the behavior of TGDPO using token-level rewards obtained from two DPO models trained with different $\beta$ values: $\beta = 0.1$ and $\beta = 0.01$. The benchmark performances of these models, along with TGDPO's performance using their respective rewards, are presented in Table 4. As expected, DPO with $\beta = 0.01$ significantly outperforms DPO with $\beta = 0.1$. However, when the token-level rewards from these models are used in TGDPO, the resulting performance is nearly identical. This finding highlights TGDPO's robustness to variations in the quality of token-level rewards, making it less dependent on the specific characteristics of the pre-trained DPO model. Such robustness further enhances TGDPO's practicality and reliability.

## 6. Conclusion

This paper enhances DPO by incorporating token-level reward guidance, which is achieved by decomposing sequence-level proximal policy optimization into a series of token-level proximal policy optimization problems. We formulate the problem of token-level proximal policy optimization with token-level reward guidance. The problem

admits a closed-form optimal token-level policy with which the corresponding token-level reward can be represented. Using the obtained token-level reward and Bradley-Terry model, we propose TGDPO, a sequence-level DPO algorithm framework with token-level reward guidance. Moreover, we introduce a practical token-level reward guidance. Extensive experiments on MT-Bench, AlpacaEval 2, and Arena-Hard demonstrate TGDPO's superiorities.

## Impact Statement

This paper enhances DPO by incorporating token-level reward guidance. This integration significantly boosts DPO's performance. Although the current evaluation concentrates on helpfulness, we believe our method would also benefit other important aspects of LLM alignment, such as safety, honesty, and fairness.

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

# A. Mathematical Derivations

## A.1. Proof of Theorem 4.1

**Theorem A.1.** *The maximum value of the sequence-level proximal policy optimization in Equation* (2) *is upper bounded by the summation from* $t = 0, 1, \ldots, T-1$ *of the maximum value of the problem:*

$$\max_{\pi_\theta} \mathbb{E}_{s_t \sim \mathcal{D}_t, a_t \sim \pi_\theta(\cdot|s_t)} \left[ r_\phi(s_t, a_t) - \beta \log \frac{\pi_\theta(a_t|s_t)}{\pi_{ref}(a_t|s_t)} \right],$$

*where* $s_t \sim \mathcal{D}_t$ *denotes that* $s_0 = x \sim \mathcal{D}$ *and* $a_p \sim \pi_\theta(\cdot|s_p)$, $p = 0, 1, \ldots, t-1$.

*Proof.* According to Section 4.1, $y \sim \prod_{t=0}^{T-1} \pi_\theta(a_t|s_t)$ in Equation (2) is equivalent to $y \sim \pi_\theta(\cdot|x)$, which is further equivalent to $s_0 = x \sim \mathcal{D}$, $a_p \sim \pi_\theta(\cdot|s_p)$, $p = 0, 1, \ldots, T-1$. Thus for the sequence-level proximal policy optimization in Equation (2),

$$\max_{\pi_\theta} \mathbb{E}_{x \sim \mathcal{D}, y \sim \prod_{t=0}^{T-1} \pi_\theta(a_t|s_t)} \left[ \sum_{t=0}^{T-1} r_\phi(s_t, a_t) - \beta \log \frac{\pi_\theta(y|x)}{\pi_{\text{ref}}(y|x)} \right]$$

$$= \max_{\pi_\theta} \mathbb{E}_{s_0 \sim \mathcal{D}, a_p \sim \pi_\theta(\cdot|s_p), p=0,1,\ldots,T-1} \left[ \sum_{t=0}^{T-1} r_\phi(s_t, a_t) - \beta \log \frac{\pi_\theta(y|x)}{\pi_{\text{ref}}(y|x)} \right]$$

$$= \max_{\pi_\theta} \mathbb{E}_{s_0 \sim \mathcal{D}, a_p \sim \pi_\theta(\cdot|s_p), p=0,1,\ldots,T-1} \left[ \sum_{t=0}^{T-1} r_\phi(s_t, a_t) - \sum_{t=0}^{T-1} \beta \log \frac{\pi_\theta(a_t|s_t)}{\pi_{\text{ref}}(a_t|s_t)} \right] \quad \text{(by Equation (6))}$$

$$= \max_{\pi_\theta} \mathbb{E}_{s_0 \sim \mathcal{D}, a_p \sim \pi_\theta(\cdot|s_p), p=0,1,\ldots,T-1} \left[ \sum_{t=0}^{T-1} \left[ r_\phi(s_t, a_t) - \beta \log \frac{\pi_\theta(a_t|s_t)}{\pi_{\text{ref}}(a_t|s_t)} \right] \right]$$

$$= \max_{\pi_\theta} \sum_{t=0}^{T-1} \mathbb{E}_{s_0 \sim \mathcal{D}, a_p \sim \pi_\theta(\cdot|s_p), p=0,1,\ldots,T-1} \left[ r_\phi(s_t, a_t) - \beta \log \frac{\pi_\theta(a_t|s_t)}{\pi_{\text{ref}}(a_t|s_t)} \right]$$

$$\leq \sum_{t=0}^{T-1} \max_{\pi_\theta} \mathbb{E}_{s_0 \sim \mathcal{D}, a_p \sim \pi_\theta(\cdot|s_p), p=0,1,\ldots,T-1} \left[ r_\phi(s_t, a_t) - \beta \log \frac{\pi_\theta(a_t|s_t)}{\pi_{\text{ref}}(a_t|s_t)} \right]$$

$$= \sum_{t=0}^{T-1} \max_{\pi_\theta} \mathbb{E}_{s_t \sim \mathcal{D}_t, a_t \sim \pi_\theta(\cdot|s_t)} \left[ r_\phi(s_t, a_t) - \beta \log \frac{\pi_\theta(a_t|s_t)}{\pi_{\text{ref}}(a_t|s_t)} \right],$$

where $s_t \sim \mathcal{D}_t$ denotes that $s_0 = x \sim \mathcal{D}$ and $a_p \sim \pi_\theta(\cdot|s_p)$, $p = 0, 1, \ldots, t-1$. This completes the proof. $\square$

## A.2. Proof of Theorem 4.3

**Theorem A.2.** *The optimal policy for the action* $a_t$ *at time step* $t$ *of the modified token-level proximal policy optimization in Equation* (11) *is:*

$$\pi_{\theta_t}(a_t|s_t) = \frac{\pi_{ref}(a_t|s_t) \exp\left( \frac{r_\phi(s_t, a_t)}{\beta f(\hat{r}(s_t, a_t))} \right)}{Z(s_t)}, \tag{22}$$

*where* $Z(s_t) = \mathbb{E}_{a_t \sim \pi_{ref}(\cdot|s_t)} \left[ \exp\left( \frac{r_\phi(s_t, a_t)}{\beta f(\hat{r}(s_t, a_t))} \right) \right]$ *is the partition function and* $s_t \sim \mathcal{D}$ *does not depend on* $\pi_{\theta_t}$. *Moreover, the token-level reward under the optimal policy is given by*

$$\frac{r_\phi(s_t, a_t)}{f(\hat{r}(s_t, a_t))} = \beta \log \frac{\pi_{\theta_t}(a_t|s_t)}{\pi_{ref}(a_t|s_t)} + \beta \log Z(s_t). \tag{23}$$

*Proof.* In Equation (11), the modified token-level proximal policy optimization is:

$$\max_{\pi_\theta} \mathbb{E}_{s_t \sim \mathcal{D}, a_t \sim \pi_\theta(\cdot|s_t)} \left[ \frac{r_\phi(s_t, a_t)}{\beta f(\hat{r}(s_t, a_t))} - \log \frac{\pi_\theta(a_t|s_t)}{\pi_{\text{ref}}(a_t|s_t)} \right]$$

$$= \max_{\pi_\theta} \mathbb{E}_{s_t \sim \mathcal{D}, a_t \sim \pi_\theta(\cdot|s_t)} \left[ \log \left( \frac{\pi_{\text{ref}}(a_t|s_t) \exp\left( \frac{r_\phi(s_t, a_t)}{\beta f(\hat{r}(s_t, a_t))} \right)}{\pi_\theta(a_t|s_t)} \right) \right]$$

$$= \max_{\pi_\theta} \mathbb{E}_{s_t \sim \mathcal{D}, a_t \sim \pi_\theta(\cdot|s_t)} \left[ \log \left( \frac{\pi_{\text{ref}}(a_t|s_t) \exp\left( \frac{r_\phi(s_t, a_t)}{\beta f(\hat{r}(s_t, a_t))} \right)}{Z(s_t)\pi_\theta(a_t|s_t)} \right) + \log Z(s_t) \right]$$

$$= \max_{\pi_\theta} \mathbb{E}_{s_t \sim \mathcal{D}, a_t \sim \pi_\theta(\cdot|s_t)} \left[ \log \left( \frac{\frac{1}{Z(s_t)}\pi_{\text{ref}}(a_t|s_t) \exp\left( \frac{r_\phi(s_t, a_t)}{\beta f(\hat{r}(s_t, a_t))} \right)}{\pi_\theta(a_t|s_t)} \right) + \log Z(s_t) \right]$$

$$= \max_{\pi_\theta} \mathbb{E}_{s_t \sim \mathcal{D}} \left[ \mathbb{E}_{a_t \sim \pi_\theta(\cdot|s_t)} \left[ \log \left( \frac{\frac{1}{Z(s_t)}\pi_{\text{ref}}(a_t|s_t) \exp\left( \frac{r_\phi(s_t, a_t)}{\beta f(\hat{r}(s_t, a_t))} \right)}{\pi_\theta(a_t|s_t)} \right) \right] + \log Z(s_t) \right]$$

$$= \max_{\pi_\theta} \mathbb{E}_{s_t \sim \mathcal{D}} \left[ -\mathbb{D}_{\text{KL}} \left[ \pi_\theta(a_t|s_t) || \frac{1}{Z(s_t)}\pi_{\text{ref}}(a_t|s_t) \exp\left( \frac{r_\phi(s_t, a_t)}{\beta f(\hat{r}(s_t, a_t))} \right) \right] + \log Z(s_t) \right] \quad (24)$$

where the partition function $Z(s_t) = \mathbb{E}_{a_t \sim \pi_{\text{ref}}(\cdot|s_t)} \left[ \exp\left( \frac{r_\phi(s_t, a_t)}{\beta f(\hat{r}(s_t, a_t))} \right) \right]$ is independent of $\pi_\theta$. Then we can define

$$\pi_{\theta_t}(a_t|s_t) = \frac{\pi_{\text{ref}}(a_t|s_t) \exp\left( \frac{r_\phi(s_t, a_t)}{\beta f(\hat{r}(s_t, a_t))} \right)}{Z(s_t)},$$

which is a valid probability distribution of action $a_t$. Furthermore in Equation (24), since $Z(s_t)$ is independent of $\pi_\theta$, the optimal policy for the action $a_t$ at time step $t$ of the modified token-level proximal policy optimization in Equation (11) can be in the form of Equation (22).

By reorganizing Equation (22), we can get the token-level reward in Equation (23). This completes the proof. $\qquad\square$

### A.3. Proof of Bradley-Terry Model with Token-Level Reward Guidance in Equation (17)

Let

$$\varphi(\pi_\theta, f, \hat{r}; x, y_w, y_l) = \sum_{t=0}^{T_w - 1} \beta f_w(\hat{r}([x, y_w^{<t}], y_w^t)) \log \frac{\pi_\theta(y_w^t|[x, y_w^{<t}])}{\pi_{\text{ref}}(y_w^t|[x, y_w^{<t}])} - \sum_{t=0}^{T_l - 1} \beta f_l(\hat{r}([x, y_l^{<t}], y_l^t)) \log \frac{\pi_\theta(y_l^t|[x, y_l^{<t}])}{\pi_{\text{ref}}(y_l^t|[x, y_l^{<t}])};$$
$$(25)$$

$$\delta(f, \hat{r}; x, y_w, y_l) = \sum_{t=0}^{T_w - 1} \beta f_w(\hat{r}([x, y_w^{<t}], y_w^t)) \log Z([x, y_w^{<t}]) - \sum_{t=0}^{T_l - 1} \beta f_l(\hat{r}([x, y_l^{<t}], y_l^t)) \log Z([x, y_l^{<t}]). \quad (26)$$

By Equation (15) and the choices of $f_w$ and $f_l$ in Section 4.3,

$$r_\phi(x, y_w)$$
$$= \sum_{t=0}^{T_w - 1} r_\phi([x, y_w^{<t}], y_w^t)$$
$$= \sum_{t=0}^{T_w - 1} \left[ \beta f_w(\hat{r}([x, y_w^{<t}], y_w^t)) \log \frac{\pi_\theta(y_w^t|[x, y_w^{<t}])}{\pi_{\text{ref}}(y_w^t|[x, y_w^{<t}])} + \beta f_w(\hat{r}([x, y_w^{<t}], y_w^t)) \log Z([x, y_w^{<t}]) \right]$$
$$= \sum_{t=0}^{T_w - 1} \beta f_w(\hat{r}([x, y_w^{<t}], y_w^t)) \log \frac{\pi_\theta(y_w^t|[x, y_w^{<t}])}{\pi_{\text{ref}}(y_w^t|[x, y_w^{<t}])} + \sum_{t=0}^{T_w - 1} \beta f_w(\hat{r}([x, y_w^{<t}], y_w^t)) \log Z([x, y_w^{<t}]).$$

Similarly,

$$r_\phi(x, y_l)$$

$$= \sum_{t=0}^{T_l-1} r_\phi([x, y_l^{<t}], y_l^t)$$

$$= \sum_{t=0}^{T_l-1} \beta f_l(\hat{r}([x, y_l^{<t}], y_l^t)) \log \frac{\pi_\theta(y_l^t|[x, y_l^{<t}])}{\pi_{\text{ref}}(y_l^t|[x, y_l^{<t}])} + \sum_{t=0}^{T_l-1} \beta f_l(\hat{r}([x, y_l^{<t}], y_l^t)) \log Z([x, y_l^{<t}].$$

In the above two equations, $T_w$ and $T_l$ are the lengths of $y_w$ and $y_l$ respectively. Thus using the notations in Equations (25) and (26) we get

$$r_\phi(x, y_w) - r_\phi(x, y_l) = \varphi(\pi_\theta, f, \hat{r}; x, y_w, y_l) + \delta(f, \hat{r}; x, y_w, y_l).$$

Then the Bradley-Terry model with the token-level reward guidance is

$$\Pr(y_w \succ y_l|x)$$

$$= \frac{\exp(r_\phi(x, y_w))}{\exp(r_\phi(x, y_w)) + \exp(r_\phi(x, y_l))}$$

$$= \frac{1}{1 + \exp(r_\phi(x, y_l) - r_\phi(x, y_w))} \tag{27}$$

$$= \sigma(r_\phi(x, y_w) - r_\phi(x, y_l))$$

$$= \sigma(\varphi(\pi_\theta, f, \hat{r}; x, y_w, y_l) + \delta(f, \hat{r}; x, y_w, y_l)).$$

### A.4. Proof of Theorem 4.4

$$\Pr(y_w \succ y_l|x) = \sigma(\varphi(\pi_\theta, f, \hat{r}; x, y_w, y_l) + \delta(f, \hat{r}; x, y_w, y_l)), \tag{28}$$

in which $\delta(f, \hat{r}; x, y_w, y_l)$ does not depend on the policy $\pi_\theta$ to be optimized, but only on $f, \hat{r}, x, y_w, y_l$ and the partition function $Z(s_t)$ (also does not depend on $\pi_\theta$, please see Theorem 4.3 in the main text). Since $\sigma(t)$ is the sigmoid function which is a strictly increasing function of $t$, we have:

**Theorem A.3.** *The function in Equation* (28) *has the same maxima and the same ascent directions as the function* $\sigma(\varphi(\pi_\theta, f, \hat{r}; x, y_w, y_l))$. *Moreover, for two policies* $\pi_{\theta_1}$ *and* $\pi_{\theta_2}$,

$$\sigma(\varphi(\pi_{\theta_1}, f, \hat{r}; x, y_w, y_l)) > \sigma(\varphi(\pi_{\theta_2}, f, \hat{r}; x, y_w, y_l)) \tag{29}$$

*if and only if*

$$\sigma(\varphi(\pi_{\theta_1}, f, \hat{r}; x, y_w, y_l) + \delta(f, \hat{r}; x, y_w, y_l))$$
$$> \sigma(\varphi(\pi_{\theta_2}, f, \hat{r}; x, y_w, y_l) + \delta(f, \hat{r}; x, y_w, y_l)). \tag{30}$$

*Proof.* Note that, $\delta(f, \hat{r}; x, y_w, y_l)$ is not dependent on the policy $\pi_\theta$, and for the sigmoid function $\sigma(t)$, it holds that $\sigma'(t) > 0$ for all $t$. Then, by the definition, $d$ is an ascent direction of the function (28) if and only if

$$d^T \nabla_{\pi_\theta} \sigma(\varphi(\pi_\theta, f, \hat{r}; x, y_w, y_l) + \delta(f, \hat{r}; x, y_w, y_l)) > 0,$$

which is equivalent to

$$d^T \sigma'(\varphi(\pi_\theta, f, \hat{r}; x, y_w, y_l) + \delta(f, \hat{r}; x, y_w, y_l)) \nabla_{\pi_\theta} \varphi(\pi_\theta, f, \hat{r}; x, y_w, y_l) > 0$$
$$\iff d^T \nabla_{\pi_\theta} \varphi(\pi_\theta, f, \hat{r}; x, y_w, y_l) > 0$$
$$\iff d^T \sigma'(\varphi(\pi_\theta, f, \hat{r}; x, y_w, y_l)) \nabla_{\pi_\theta} \varphi(\pi_\theta, f, \hat{r}; x, y_w, y_l) > 0$$
$$\iff d^T \nabla_{\pi_\theta} \sigma(\varphi(\pi_\theta, f, \hat{r}; x, y_w, y_l)) > 0.$$

Hence the function (28) has the same ascent directions as the function $\sigma\left(\varphi(\pi_\theta, f, \hat{r}; x, y_w, y_l)\right)$. Similarly,

$$
\begin{aligned}
&\nabla_{\pi_\theta} \sigma\left(\varphi(\pi_\theta, f, \hat{r}; x, y_w, y_l) + \delta(f, \hat{r}; x, y_w, y_l)\right) = 0 \\
&\iff \sigma'\left(\varphi(\pi_\theta, f, \hat{r}; x, y_w, y_l) + \delta(f, \hat{r}; x, y_w, y_l)\right) \nabla_{\pi_\theta} \varphi(\pi_\theta, f, \hat{r}; x, y_w, y_l) = 0 \\
&\iff \nabla_{\pi_\theta} \varphi(\pi_\theta, f, \hat{r}; x, y_w, y_l) = 0 \\
&\iff \sigma'\left(\varphi(\pi_\theta, f, \hat{r}; x, y_w, y_l)\right) \nabla_{\pi_\theta} \varphi(\pi_\theta, f, \hat{r}; x, y_w, y_l) = 0 \\
&\iff \nabla_{\pi_\theta} \sigma\left(\varphi(\pi_\theta, f, \hat{r}; x, y_w, y_l)\right) = 0.
\end{aligned}
$$

Thus, the function in Equation (28) has the same maxima and the same ascent directions as the function $\sigma\left(\varphi(\pi_\theta, f, \hat{r}; x, y_w, y_l)\right)$.

Next, since $\sigma(t)$ is strictly increasing, for inequality Equation (29) we have

$$
\begin{aligned}
&\sigma\left(\varphi(\pi_{\theta_1}, f, \hat{r}; x, y_w, y_l)\right) > \sigma\left(\varphi(\pi_{\theta_2}, f, \hat{r}; x, y_w, y_l)\right) \\
&\iff \varphi(\pi_{\theta_1}, f, \hat{r}; x, y_w, y_l) > \varphi(\pi_{\theta_2}, f, \hat{r}; x, y_w, y_l) \\
&\iff \varphi(\pi_{\theta_1}, f, \hat{r}; x, y_w, y_l) + \delta(f, \hat{r}; x, y_w, y_l) > \varphi(\pi_{\theta_2}, f, \hat{r}; x, y_w, y_l) + \delta(f, \hat{r}; x, y_w, y_l) \\
&\iff \sigma\left(\varphi(\pi_{\theta_1}, f, \hat{r}; x, y_w, y_l) + \delta(f, \hat{r}; x, y_w, y_l)\right) > \sigma\left(\varphi(\pi_{\theta_2}, f, \hat{r}; x, y_w, y_l) + \delta(f, \hat{r}; x, y_w, y_l)\right).
\end{aligned}
$$

$\square$

For easy understanding of Theorem A.3, we simplify in the sequel all notations independent of $\pi_\theta$ to be optimized, then the Bradley-Terry preference model in Equation (28) is $\Pr(y_w \succ y_l | x) = \sigma(\varphi(\pi_\theta) + \delta)$, and Theorem A.3 is exactly as:

**Theorem A.4.** *For the policy $\pi_\theta$, the function $\sigma(\varphi(\pi_\theta) + \delta)$ has the same maxima and ascent directions as the function $\sigma(\varphi(\pi_\theta))$, here $\sigma(t)$ is the sigmoid function.*

*Proof.* Note that the sigmoid function $\sigma(t)$ is strictly increasing, meaning that for any real numbers $a$ and $b$, $a \geq b$ if and only if $\sigma(a) \geq \sigma(b)$. Thus, if $\pi_\theta^*$ is a maximal solution of $\sigma(\varphi(\pi_\theta) + \delta)$, then by the definition, there exists a neighborhood $\mathcal{N}$ of $\pi_\theta^*$ such that $\forall \pi_\theta \in \mathcal{N}, \sigma(\varphi(\pi_\theta^*) + \delta) \geq \sigma(\varphi(\pi_\theta) + \delta)$. So $\varphi(\pi_\theta^*) + \delta \geq \varphi(\pi_\theta) + \delta$, and $\varphi(\pi_\theta^*) \geq \varphi(\pi_\theta)$. This leads to $\sigma(\varphi(\pi_\theta^*)) \geq \sigma(\varphi(\pi_\theta))$, meaning that $\pi_\theta^*$ is also a maximal solution of $\sigma(\varphi(\pi_\theta))$. The converse can be proved similarly.

Next, $d$ is an ascent direction of the function $\sigma(\varphi(\pi_\theta) + \delta)$ if and only if

$$
d^T \nabla_{\pi_\theta} \sigma(\varphi(\pi_\theta) + \delta) = \sigma'(\varphi(\pi_\theta) + \delta) d^T \nabla_{\pi_\theta} \varphi(\pi_\theta) > 0,
$$

which is equivalent to

$$
\begin{aligned}
&d^T \nabla_{\pi_\theta} \varphi(\pi_\theta) > 0 \\
&\iff \sigma'(\varphi(\pi_\theta)) d^T \nabla_{\pi_\theta} \varphi(\pi_\theta) > 0 \\
&\iff d^T \sigma'(\varphi(\pi_\theta)) \nabla_{\pi_\theta} \varphi(\pi_\theta) > 0 \\
&\iff d^T \nabla_{\pi_\theta} \sigma(\varphi(\pi_\theta)) > 0.
\end{aligned}
$$

Hence the function $\sigma(\varphi(\pi_\theta) + \delta)$ has the same ascent directions as the function $\sigma(\varphi(\pi_\theta))$ w.r.t. $\pi_\theta$.

Further, the sigmoid function is strictly increasing, it does not change the order of values. $\square$

# B. More Experiment Results

## B.1. Additional Baseline Comparison

Below we supplement the result of TDPO (Zeng et al., 2024) as an additional baseline for the experiment in Table 1 of the main paper. The additional result is demonstrated in Table 5. It can be seen that the performance of TDPO is very close to that of DPO. Our TGDPO, on the other hand, outperforms DPO by a large margin. Our TGDPO aims to leverage an existing token-level reward to guide DPO training at the token level. Whereas, TDPO (Zeng et al., 2024) aims to enhance the regulation of KL-divergence by incorporating a forward KL-divergence for each token to the DPO objective. It is not guided by a token-level reward.

*Table 5.* Experiment results on AlpacaEval 2 (Li et al., 2023), Arena-Hard (Li et al., 2024), and MT-Bench (Zheng et al., 2023) benchmarks.

| Method | Llama3-8B-Instruct PairRM | | | | Llama3-8B-Instruct ArmoRM | | | |
| | AlpacaEval 2 | Arena-Hard | MT-Bench | | AlpacaEval 2 | Arena-Hard | MT-Bench | |
| | Win Rate (%) | Win Rate (%) | Score | Win Rate(%) | Win Rate (%) | Win Rate (%) | Score | Win Rate(%) |
| --- | --- | --- | --- | --- | --- | --- | --- | --- |
| SFT | 30.6 | 21.4 | 7.9 | 27.5 | 30.6 | 21.4 | 7.9 | 27.5 |
| DPO | 41.7 | 30.4 | **8.0** | 37.5 | 40.8 | 36.2 | **8.2** | **46.3** |
| TDPO | 40.7 | 30.2 | **8.0** | 39.0 | 41.3 | 36.7 | 8.0 | 42.5 |
| SimPO | 39.8 | 28.7 | 7.8 | 32.5 | 37.0 | 28.1 | 7.8 | 42.5 |
| TGDPO | **43.9** | **34.3** | **8.0** | **41.9** | **42.5** | **40.5** | 7.9 | 45.0 |
| Method | Llama3.2-3B-Instruct ArmoRM | | | | Gemma2-2B-it ArmoRM | | | |
| | AlpacaEval 2 | Arena-Hard | MT-Bench | | AlpacaEval 2 | Arena-Hard | MT-Bench | |
| | Win Rate (%) | Win Rate (%) | Score | Win Rate (%) | Win Rate (%) | Win Rate (%) | Score | Win Rate (%) |
| SFT | 23.8 | 17.1 | 7.0 | 16.3 | 32.8 | 20.1 | 7.9 | 37.5 |
| DPO | 29.6 | 23.2 | 7.9 | 29.4 | 40.8 | 26.4 | 8.0 | 43.1 |
| TDPO | 30.3 | 23.1 | 7.8 | 30.0 | 41.5 | 27.0 | 8.0 | 40.0 |
| SimPO | 26.2 | 22.6 | 7.4 | 15.7 | 34.8 | 21.1 | 7.8 | 40.0 |
| TGDPO | **35.8** | **25.4** | **8.1** | **36.9** | **43.0** | **30.7** | **8.1** | **46.9** |

*Table 6.* Experiment results on SFT models on AlpacaEval 2 (Li et al., 2023), Arena-Hard (Li et al., 2024), and MT-Bench (Zheng et al., 2023) benchmarks.

| Method | Llama3-8B-SFT-Mixture Ultrafeedback | | | |
| | AlpacaEval 2 | Arena-Hard | MT-Bench | |
| | Win Rate (%) | Win Rate (%) | Score | Win Rate(%) |
| --- | --- | --- | --- | --- |
| SFT | 5.0 | 6.2 | 7.6 | 16.3 |
| DPO | 9.9 | 10.2 | **7.8** | 19.5 |
| TDPO | 11.0 | 11.7 | 7.5 | 15.7 |
| SimPO | 16.4 | 21.4 | **7.8** | 27.5 |
| TGDPO w/ DPO's token reward | 12.8 | 13.8 | 7.7 | 20.0 |
| TGDPO w/ SimPO's token reward | **26.9** | **25.3** | 7.6 | **31.9** |

## B.2. Experiment on SFT Models

In this section, we conduct experiments starting from SFT models. Specifically, we use the open-source SFT model Llama3-8B-SFT-Mixture from OpenRLHF (Hu et al., 2024). Llama3-8B-SFT-Mixture is trained using diverse, high-quality, open-source datasets by SFT and has not been trained by RLHF. Following (Meng et al., 2024), we conduct preference optimization on the UltraFeedback dataset (Cui et al., 2024) using the SFT model as the starting point.

The experiment results on SFT models on AlpacaEval 2 (Li et al., 2023), Arena-Hard (Li et al., 2024), and MT-Bench (Zheng et al., 2023) are shown in Table 6. Our TGDPO can leverage the token-level rewards from DPO or SimPO and outperforms them correspondingly. Specifically, TGDPO using SimPO's token-level reward achieves much better performance than all baseline methods. It achieves win rate gains of 10.5 on AlpacaEval 2, 4.4 on MT-Bench, and 3.9 on Arena-Hard compared to best-performing baselines.

## C. More Discussions on Closely Related Work

Our work proposes a framework for incorporating existing token-level rewards explicitly into the loss function of DPO, to guide optimizing policy at a fine-grained level. This is a challenging task since DPO's reward function is explicitly expressed through the policy being optimized. Especially, a key theoretical challenge in deriving the computable loss function in Equation (20) is the elimination of the partition functions, which is addressed in Theorem 4.4 or Theorem A.3 or Theorem A.4.

## C.1. Closely Related Work

*Table 7.* Per-instance losses of closely related direct optimization methods.

| Method | Per-Instance Loss |
|---|---|
| **TDPO** (Zeng et al., 2024) | $\sigma(u(x, y_w, y_l) - \delta(x, y_w, y_l))$, 

 where $u(x, y_w, y_l) = \beta \log \frac{\pi_\theta(y_w\|x)}{\pi_{\text{ref}}(y_w\|x)} - \beta \log \frac{\pi_\theta(y_l\|x)}{\pi_{\text{ref}}(y_l\|x)}$, 

 $\delta(x, y_w, y_l)) = \beta D_{\text{SeqKL}}(x, y_l; \pi_{\text{ref}}\|\|\pi_\theta) - \beta D_{\text{SeqKL}}(x, y_w; \pi_{\text{ref}}\|\|\pi_\theta)$. |
| **Yang et al. (2024)** | $\sigma\left(C\mathbb{E}_{t\sim\text{Cat}(\{\gamma^t\})}\left[\log \frac{\pi_\theta(a_t^w\|s_t^w)}{\pi_{\text{ref}}(a_t^w\|s_t^w)} - \log \frac{\pi_\theta(a_t^l\|s_t^l)}{\pi_{\text{ref}}(a_t^l\|s_t^l)}\right]\right)$ |
| **D²PO** (Shao et al., 2025) | $\sigma\left(\sum_{t=0}^{T_w} \gamma^t \beta \log \frac{\pi_\theta(y_w^t\|x, y_w^{<t})}{\pi_{\text{ref}}(y_w^t\|x, y_w^{<t})} - \sum_{t=0}^{T_l} \gamma^t \beta \log \frac{\pi_\theta(y_l^t\|x, y_l^{<t})}{\pi_{\text{ref}}(y_l^t\|x, y_l^{<t})}\right)$ |
| **TGDPO** (ours) | $\sigma\left(\sum_{t=0}^{T_w-1} \beta f_w(\hat{r}([x, y_w^{\leq t}], y_w^t)) \log \frac{\pi_\theta(y_w^t\|[x, y_w^{<t}])}{\pi_{\text{ref}}(y_w^t\|[x, y_w^{<t}])} - \sum_{t=0}^{T_l-1} \beta f_l(\hat{r}([x, y_l^{\leq t}], y_l^t)) \log \frac{\pi_\theta(y_l^t\|[x, y_l^{<t}])}{\pi_{\text{ref}}(y_l^t\|[x, y_l^{<t}])}\right)$ |

Several direct preference optimization methods also perform in a token-level manner. We derive our modified token-level reward beginning from Equation (9), which is similar to those in Zeng et al. (2024) and Yang et al. (2024). However, the obtained final per-instance losses are different. These per-instance losses are listed in Table 7 for comparisons. From the table, it is obvious that our TGDPO explicitly incorporates existing token-level rewards into the per-instance loss for guiding DPO. While, TDPO (Zeng et al., 2024) constrains each token with forward KL-divergence, and fine-tunes pre-trained LLMs from the token level to enhance the regulation of KL-divergence. Additionally, Yang et al. (2024) and D²PO (Shao et al., 2025) focus on earlier tokens of sequential generation for their tasks, by posing temporal decay parameters.

Moreover, in the derivation of our TGDPO, the partition function $Z(\cdot)$ is not dependent on the policy to be optimized, and it can be eliminated from the loss function by using our developed Theorem 4.4 or Theorem A.3 or Theorem A.4, which are new and powerful. While, in TDPO (Zeng et al., 2024) the partition function is kept in their loss function and is changed to the forward KL-divergence. Yang et al. (2024) managed to eliminate the partition function from their loss function using the lower bounding approach. The method in D²PO (Shao et al., 2025) does not involve a partition function, since it is derived from the KL-constrained RL objective under the maximum entropy RL setting.

## C.2. Recovering Several Direct Preference Optimization Methods

In Section 4.2, we mentioned that the loss function $\mathcal{L}_{\text{TGDPO}}(\pi_\theta)$ in Equation (20) provides a framework of direct preference optimization with token-level reward guidance. With an appropriate choice of $f(\cdot)$, this framework can recover several known direct preference optimization methods. For example, if we take $f_w \equiv f_l \equiv 1$, then Equation (20) is the loss function of DPO (Rafailov et al., 2023). In the following, we give some other examples. **It must be noted that these known preference optimization methods have their respective motivations. We only want to demonstrate that our proposed framework is reasonable by recovering them here.**

Note that, our per-instance loss of $\mathcal{L}_{\text{TGDPO}}(\pi_\theta)$ is

$$\mathcal{L}_{\text{TGDPO\_P}}(\pi_\theta) = \sigma\left(\sum_{t=0}^{T_w-1} \beta f_w(\hat{r}([x, y_w^{\leq t}], y_w^t)) \log \frac{\pi_\theta(y_w^t|[x, y_w^{<t}])}{\pi_{\text{ref}}(y_w^t|[x, y_w^{\leq t}])} - \sum_{t=0}^{T_l-1} \beta f_l(\hat{r}([x, y_l^{\leq t}], y_l^t)) \log \frac{\pi_\theta(y_l^t|[x, y_l^{<t}])}{\pi_{\text{ref}}(y_l^t|[x, y_l^{<t}])}\right).$$

1. Recovering the per-instance loss of SimPO (Meng et al., 2024): By setting $f_w(\hat{r}([x, y_w^{\leq t}], y_w^t)) = \frac{1}{|y_w|}$ and

$f_l(\hat{r}([x, y_l^{<t}], y_l^t)) = \frac{1}{|y_w|}$, we get

$$\mathcal{L}_{\text{TGDPO\_P}}(\pi_\theta) = \sigma\left(\sum_{t=0}^{T_w-1} \frac{\beta}{|y_w|} \log \frac{\pi_\theta(y_w^t|[x, y_w^{<t}])}{\pi_{\text{ref}}(y_w^t|[x, y_w^{<t}])} - \sum_{t=0}^{T_l-1} \frac{\beta}{|y_l|} \log \frac{\pi_\theta(y_l^t|[x, y_l^{<t}])}{\pi_{\text{ref}}(y_l^t|[x, y_l^{<t}])}\right)$$

$$= \sigma\left(\frac{\beta}{|y_w|} \sum_{t=0}^{T_w-1} \log \frac{\pi_\theta(y_w^t|[x, y_w^{<t}])}{\pi_{\text{ref}}(y_w^t|[x, y_w^{<t}])} - \frac{\beta}{|y_l|} \sum_{t=0}^{T_l-1} \log \frac{\pi_\theta(y_l^t|[x, y_l^{<t}])}{\pi_{\text{ref}}(y_l^t|[x, y_l^{<t}])}\right)$$

$$= \sigma\left(\frac{\beta}{|y_w|} \log \frac{\pi_\theta(y_w|x)}{\pi_{\text{ref}}(y_w|x)} - \frac{\beta}{|y_l|} \log \frac{\pi_\theta(y_l|x)}{\pi_{\text{ref}}(y_l|x)}\right)$$

$$= \sigma\left(\frac{\beta}{|y_w|} \log \pi_\theta(y_w|x) - \frac{\beta}{|y_l|} \log \pi_\theta(y_l|x) + \left(\frac{\beta}{|y_l|} \log \pi_{\text{ref}}(y_l|x) - \frac{\beta}{|y_w|} \log \pi_{\text{ref}}(y_w|x)\right)\right).$$

Furthermore, by Theorem 4.4 or Theorem A.3 or Theorem A.4, introducing a constant into the above function does not change where the function is maximized. Hence we get

$$\mathcal{L}_{\text{TGDPO\_P}}(\pi_\theta) = \sigma\left(\frac{\beta}{|y_w|} \log \pi_\theta(y_w|x) - \frac{\beta}{|y_l|} \log \pi_\theta(y_l|x) - \gamma\right),$$

which is exactly the per-instance loss of SimPO.

2. Recovering the per-instance loss of R-DPO (Park et al., 2024): By setting $f_w(\hat{r}([x, y_w^{<t}], y_w^t)) = f_l(\hat{r}([x, y_l^{<t}], y_l^t)) \equiv 1$, we get

$$\mathcal{L}_{\text{TGDPO\_P}}(\pi_\theta) = \sigma\left(\sum_{t=0}^{T_w-1} \beta \log \frac{\pi_\theta(y_w^t|[x, y_w^{<t}])}{\pi_{\text{ref}}(y_w^t|[x, y_w^{<t}])} - \sum_{t=0}^{T_l-1} \beta \log \frac{\pi_\theta(y_l^t|[x, y_l^{<t}])}{\pi_{\text{ref}}(y_l^t|[x, y_l^{<t}])}\right)$$

$$= \sigma\left(\beta \log \frac{\pi_\theta(y_w|x)}{\pi_{\text{ref}}(y_w|x)} - \beta \log \frac{\pi_\theta(y_l|x)}{\pi_{\text{ref}}(y_l|x)}\right).$$

Furthermore, since $\alpha|y_w| - \alpha|y_l|$ does not depend on the policy $\pi_\theta$, by Theorem 4.4 or Theorem A.3 or Theorem A.4, introducing it into the above function does not change where the function is maximized. Hence we get

$$\mathcal{L}_{\text{TGDPO\_P}}(\pi_\theta) = \sigma\left(\beta \log \frac{\pi_\theta(y_w|x)}{\pi_{\text{ref}}(y_w|x)} - \beta \log \frac{\pi_\theta(y_l|x)}{\pi_{\text{ref}}(y_l|x)} + (\alpha|y_w| - \alpha|y_l|)\right).$$

which is exactly the per-instance loss of R-DPO.

3. Recovering the per-instance loss of D²PO (Shao et al., 2025): By setting $f_w(\hat{r}([x, y_w^{<t}], y_w^t)) = f_l(\hat{r}([x, y_l^{<t}], y_l^t)) = \gamma^t$, we immediately get

$$\mathcal{L}_{\text{TGDPO\_P}}(\pi_\theta) = \sigma\left(\sum_{t=0}^{T_w-1} \beta\gamma^t \log \frac{\pi_\theta(y_w^t|[x, y_w^{<t}])}{\pi_{\text{ref}}(y_w^t|[x, y_w^{<t}])} - \sum_{t=0}^{T_l-1} \beta\gamma^t \log \frac{\pi_\theta(y_l^t|[x, y_l^{<t}])}{\pi_{\text{ref}}(y_l^t|[x, y_l^{<t}])}\right),$$

which is exactly the per-instance loss of D²PO.

# D. Implementation Details

## D.1. Hyperparameter Settings

Following (Meng et al., 2024), we use a consistent batch size of 128 and train all methods for 1 epoch in all settings. The AdamW optimizer (Loshchilov & Hutter, 2019) is used. The max sequence length is set to be 2048 and a cosine learning rate schedule with 10% warm-up steps is used. The hyperparameters for each method are grid-searched and are shown in Table 8 for DPO, Table 9 for SimPO, Table 10 for our TGDPO correspondingly. TDPO in Appendix B.1 uses the same hyperparameters as DPO with an additional KL-penalty scale of 0.01. The training is conducted using 8 A100 GPUs.

*Table 8.* The hyperparameters of DPO for each training setting.

| Setting | $\beta$ | learning rate |
|---|---|---|
| **Llama3-8B-Instruct PairRM** | 0.01 | 7e-7 |
| **Llama3-8B-Instruct ArmoRM** | 0.01 | 7e-7 |
| **Llama3.2-3B-Instruct ArmoRM** | 0.1 | 7e-7 |
| **Gemma2-2B-it ArmoRM** | 0.1 | 5e-7 |
| **Llama3-8B-SFT-Mixture Ultrafeedback** | 0.1 | 5e-7 |

*Table 9.* The hyperparameters of SimPO for each training setting.

| Setting | $\beta$ | $\gamma$ | learning rate |
|---|---|---|---|
| **Llama3-8B-Instruct PairRM** | 2.5 | 1.4 | 1e-6 |
| **Llama3-8B-Instruct ArmoRM** | 10 | 3.0 | 1e-6 |
| **Llama3.2-3B-Instruct ArmoRM** | 10 | 3.0 | 1e-6 |
| **Gemma2-2B-it ArmoRM** | 20 | 2.0 | 5e-7 |
| **Llama3-8B-SFT-Mixture Ultrafeedback** | 2.5 | 0.5 | 5e-7 |

*Table 10.* The hyperparameters of TGDPO for each training setting.

| Setting | $\beta$ for $\hat{r}(s_t, a_t)$ | $\gamma$ for $\hat{r}(s_t, a_t)$ | $\beta$ | $\alpha$ | learning rate |
|---|---|---|---|---|---|
| **Llama3-8B-Instruct PairRM** | 0.01 | - | 0.1 | 0.5 | 7e-7 |
| **Llama3-8B-Instruct ArmoRM** | 0.01 | - | 0.1 | 0.2 | 7e-7 |
| **Llama3.2-3B-Instruct ArmoRM** | 0.1 | - | 0.1 | 2.0 | 7e-7 |
| **Gemma2-2B-it ArmoRM** | 0.1 | - | 0.1 | 0.5 | 5e-7 |
| **Llama3-8B-SFT-Mixture Ultrafeedback w/ DPO's token reward** | 0.1 | - | 0.1 | 0.2 | 7e-7 |
| **Llama3-8B-SFT-Mixture Ultrafeedback w/ SimPO's token reward** | 2.5 | 0.5 | 0.01 | 1.2 | 7e-7 |

## D.2. Benchmark Details

Following (Meng et al., 2024), we use a decoding temperature of 0.9 for the Llama models and a decoding temperature of 0.5 for the Gemma models for AlpacaEval 2. For Arena-Hard, we use the default greedy decoding for all models. We use the latest GPT-4o-2024-11-20 as the judge model for AlpacaEval 2 and Arena-Hard. We follow the official default configurations on MT-Bench.

