# OpenReview forum: "TGDPO: Harnessing Token-Level Reward Guidance for Enhancing Direct Preference Optimization"
_ICML.cc/2025/Conference — ICML 2025 poster_

### Official Review · Reviewer_aioX · 2025-02-15

**Overall Recommendation:** 2

**Summary:**

Recent work in RLHF has revealed the benefits of utilizing fine-grained rewards. The combination of token-level guidance with DPO, however, remains to be explored.
To address this challenge, this paper decomposes the sequence-level RL formulation in the original DPO derivation as a sequence of token-level RL problem, from which closed-form solution for the optimal token-level policy and reward can be derived.
From this, this paper derives a loss with token-level reward guidance for DPO and proposes a practical reward guidance based on the induced DPO reward.
This formulation enables different tokens to exhibit varying degrees of deviation from reference policy.

**Claims And Evidence:**

Yes, the empirical performance validates the proposed method.

**Essential References Not Discussed:**

See the Weaknesses section

**Experimental Designs Or Analyses:**

Yes, tested on standard RLHF benchmarks

**Methods And Evaluation Criteria:**

Yes.

**Other Comments Or Suggestions:**

1. "sentence-level reward" is not accurate, what the authors refer to should be termed as bandit/{sequence, trajectory, response}-level reward, since each response may contain multiple sentences, for example, in the summarization task. Ditto all presence of "sentence".

2. Most of the appearance of "proximal policy optimization problem"/"PPO problem" should be changed to "KL-regularized policy optimization"/"KL-regularized RL"/"KL-regularized control".

**Other Strengths And Weaknesses:**

## Strengths
1. Dense/token-level reward is a promising direction for RLHF to improve upon the classical bandit/sequence-level formulation. The combination of this direction with DPO is particularly less cultivated.

2. The proposed method shows competitive performance gain.



## Weaknesses

1. The idea of "token-level reward guidance + DPO"  has been explored, e.g., in [1, 2]. Specially, Equ. (8) in this paper is almost identical to Equ. (12) in [1]. The authors need to properly discuss the connection and novelty compared to these prior works.

2. Some of the basic terms/concepts are confused with others, see the next "Suggestions" section.

3. Please check the Questions section for a potential mistakes on the Method section and several unclear parts.

***
[1] A Dense Reward View on Aligning Text-to-Image Diffusion with Preference. In ICML 2024.

[2] Earlier Tokens Contribute More: Learning Direct Preference Optimization From Temporal Decay Perspective. In ICLR 2025.

**Questions For Authors:**

1. Could you explain what is "sentence-level proximal policy optimization with token-level reward guidance"? In particular, why is PPO sentence-level given that the reward is token-level?

2. In Assumption 4.2:
  - What is "the **corresponding** dense token-level reward"? Why does the reward need to be learnt to correspond to the policy?
  - What is the different between $\hat r$ and $r_\phi$? Why couldn't we optimize $\pi_\theta$ against $\hat r$?
  - What is the requirement of the function $f$? The current description looks like a constant function of 1 is also valid, which will trivialize Equ. (10). For the proof of Equ. (15), I believe we need the stronger assumption of $\epsilon \times M \times T_w \approx 0$.

3. What is the benefit of introducing $f(\hat r)$ in Equ. (10)? See the following question for a problem with the current specification of $f$.

4. L286: "likely to make this token as a dispreferred one" --- this is incorrect. The current loss will only make the optimization strength of those token relatively smaller, rather than minimizing their likelihood, since Assumption 4.2 says $|f(u)| \geq 1 - \epsilon > 0$ for small $\epsilon$. Ditto L299. With this, the author may need to revise the claim in L406-408 about "reduces conflicts".

**Relation To Broader Scientific Literature:**

See the Strength section.

**Theoretical Claims:**

Yes

---

> ### Author Rebuttal · Authors · 2025-04-01
>
> **Q1 Comparison with existing works**
>
> All [1], [2] and ours are based on the assumption of dense token-level reward to derive respective DPO loss functions. Eq. (8) in our work seems similar to Eq. (12) in [1], but our Eq. (8) is not solvable since $s_t\sim\mathcal{D}\_t$ is dependent on the policy $\pi\_{\theta}$ to be optimized. This has been demonstrated in line 219 (left) - line 168 (right) of the main text. In the final version we will give citations, discuss connections with [1, 2] and the differences highlighted as follows:
>
> **Motivation Differences:**  We intend to incorporate a learned token-level reward explicitly into DPO. While, they only focus on asigning more weight on earlier tokens.
>
> **Loss Function Differences:** Our work explicitly incorporate an existing token-level reward in the loss function for guiding DPO. [1] uses the same loss as DPO and modifies the sampling probability for different diffusion timesteps. [2] adds a temporal decay factor for each token in the DPO loss. Their loss functions do not leverage token-level reward.
>
> **Method Differences:** Integrating token-level reward in DPO's loss function explicitly presents distinctive difficulties, especially in the derivation. Thanks to all reviewers' comments, we have proposed a new approach for canceling partition functions in the Bradley-Terry model, please see **Q7**.
>
> [1] A Dense Reward View on Aligning Text-to-Image Diffusion with Preference
>
> [2] Earlier Tokens Contribute More: Learning Direct Preference Optimization From Temporal Decay Perspective
>
> **Q2 Accurate word choice**
>
> We will change to use "response-level reward" and "KL-regularized policy optimization" to improve accuracy.
>
> **Q3 Explain "sentence-level PPO with token-level reward guidance"**
>
> A sequence of dense token-level reward $r_{\phi}(s_t, a_t)$ is obtained before PPO, and then it can guide PPO finetuning in a more fine-grained way.
>
> **Q4 What is the corresponding dense token-level reward**
>
> Actually, $\pi_{\hat\theta}$ does not necessarily relate to $\hat{r}$ in Assumption 4.2. $\pi_{\hat\theta}$ is only used for sampling $s_t$ in Equation (10). Now, the first sentence of Assumption 4.2 is modified as: Suppose we have learned a dense token-level reward $\hat{r}$ using some effective learning approach. In the revision, we will improve the description and notation to reduce misunderstanding.
>
> **Q5 Difference between $\hat{r}$ and $r_{\phi}$**
>
> $\hat{r}$ is an existing dense token-level reward, used for shaping the reward $r_{\phi}$ in Equ. (10). Following the derivation of DPO, $r_{\phi}$ is expressed with $\hat r$, $\pi_{\theta}$ and $\pi_{\text{ref}}$ by solving Equ. (10). And finally, $\hat{r}$ will appear in Equ. (16) of our loss function, but $r_{\phi}$ does not.
>
> If we optimize $\pi_\theta$ against $\hat r$, then Equ. (10) becomes
> $$
> \max_{\pi_{{\bf\theta}}} \mathbb{E}\_{s_t\sim\hat{\mathcal{D}}\_t, a_t\sim \pi_{\theta} (\cdot|s_t) } \left[{r_{\phi}(s_t, a_t)} - {\beta f(\hat{r}(s_t, a_t))}\log \frac{ \pi_{\theta} (a_t|s_t)}{\pi_{\text{ref}}(a_t|s_t)}\right].
> $$
> We tried and found it is difficult to solve the problem due to the expectation on the product term.
>
> **Q6 Requirement for $f$**
>
> In Assumption 4.2, the $f$ satisfying $f(0)=1$ and $|f(u)-1|\le\varepsilon$ is sufficient for deriving our TGDPO. Concrete $f(u)$ can be chosen by developers personally. As for us, Equ. (17) is adopted in our experiments. If $f(u)\equiv 1$ then our TGDPO degenerates to DPO, which is not interesting.
>
>  **Q7 Strong assumption for Equation 15**
>
> Thank you very much for the comment! Now the assumptions have been completely removed, the approach is novel. The theorem and proof can be seen in response to **Q3 of Reviewer F88J**. We will provide revisions in the final version.
>
> **Q8 Benefit of introducing $f(\hat r)$ in Equ. (10)**
>
> The benefit of introducing $f(\hat r)$ in Equ. (10) is in Equ. (16) of the loss function.
> We adopt Equ. (17) in our experiments. Then take win response $y_w$ for an example, the other is similar. In Equ. (16),
> $$
> f(\hat{r}([x, y_w^{<t}], y_w^t))   \log\frac {\pi_{\theta} (y_w^t|[x, y_w^{<t}])}{\pi_{\text{ref}}(y_w^t|[x, y_w^{<t}])} = (1 + \alpha \hat{r}([x, y_w^{<t}], y_w^t)) \log\frac {\pi_{\theta} (y_w^t|[x, y_w^{<t}])}{\pi_{\text{ref}}(y_w^t|[x, y_w^{<t}])}.
> $$
> For $\hat{r}([x, y_w^{<t}], y_w^t)>0$, we emphasize more on this token, otherwise less or keep the weight unchanged as 1.
>
> **Q9 Improper description in L286, L406-408**
>
> Thank you very much for pointing out the issues. We modify them as:
> 1) L283-286 (right): "Then this token is optimized with less strength  during the optimization of the loss function  $\mathcal{L}\_{\text{TGDPO}}(\pi_{\theta})$, since $f(\hat{r}([x, y_w^{<t}], y_w^t)) <1$."
> 2) L296-299 (right): Similar to point 1.
> 3. L406-408 (left): "distinguish preferred tokens in chosen samples and dispreferred tokens in rejected
> ones, TGDPO takes care of them and enables ..."

---

### Official Review · Reviewer_F88J · 2025-02-23

**Overall Recommendation:** 2

**Summary:**

This paper proposes a method that integrates Direct Preference Optimization (DPO) with token-level rewards. The paper first provide an upper bound for rewards in sentence-level LLM generation by decomposing the problem into a series of token-level reward maximization tasks. Building on this foundation, the paper adapts the DPO method to solve these token-level reward-maximization problem. Experiments were conducted on three models and evaluated across three benchmarks. The results demonstrate that the proposed method outperforms baseline approaches.

**Claims And Evidence:**

In this paper it is claimed that the proposed TGDPO performs better than baselines like DPO and SimPO. This is adequately supported by experiments in Section 5 of this paper.

**Essential References Not Discussed:**

I don't see a missing of significant related works.

**Experimental Designs Or Analyses:**

Extensive experiments are conducted, and the results looks convincing to me.

**Methods And Evaluation Criteria:**

The paper mainly focuses on RLHF problem. The evaluation benchmark involved, namely AlpacaEval, MT-Bench and Arena-Hard, are all widely used benchmarks for alignment evaluation.

**Other Comments Or Suggestions:**

N/A

**Other Strengths And Weaknesses:**

N/A

**Questions For Authors:**

See above

**Relation To Broader Scientific Literature:**

This paper adapts DPO to token-level reward and the relation with previous works are mostly clearly stated

**Theoretical Claims:**

The concerns regarding methodology are listed below

1.While Theorem 4.1 appears to be correct, its intended message is unclear. The objective in equation (8) can certainly serve as an upper bound for equation (2), but there is no guarantee that a policy maximizing (8) will also maximize (2). To be more specific, it looks like the objective in equation (8) is to myopically optimize the reward of the currect step without considering further states. However, in RL, the goal is to maximize the expectation of **total** future rewards (i.e., Q-function).  policy that optimizes one-step rewards might lead the agent into unfavorable states, ultimately hindering its ability to achieve high cumulative rewards.

2. The introduction of function $f$ in Assumption 4.2 appears insufficiently motivated. Could the authors provide further explanation and justification for its inclusion?

3. In equation 15, the $\approx$ is obtained by assuming $Z(x, y_w^{<t}) \approx Z(x, y_l^{<t})$.However, this assumption seems questionable. Even if $y_w$ and $y_l$ are both generated on-policy, since there is randomness during sample and the generated response is long, even some slight variation at the beginning might leads to significantly different prefix $y^{<t}$. Therefore it is not appropriate to make such assumption.

4. The equation right above equation (17) gives a way to distribute sentence level reward to token-level reward. However, such distribution is not well-motivated. In fact, [1] gives the exact reward distribution as $r(s_t,a_t) = \beta \log \pi_{\theta} / \pi_{\text{ref}} + V(s_{t}) - V(s_{t+1})$. Therefore, the reward distribution is valid only when $V(s_{t}) = V(s_{t+1})$, which is not established in the paper.

[1] Rafailov, Rafael, et al. "From $ r $ to $ q^* $: Your language model is secretly a q-function." arXiv preprint arXiv:2404.12358 (2024).

---

> ### Author Rebuttal · Authors · 2025-04-01
>
> **Q1 Clarification regarding Equ. 8**
>
> By [1] (Thanks to  Reviewer  **aioX**), Equ. (8) has connections to an approximation approach common in prior RL works [2, 3, 4]. [1] pointed out this by providing 5 papers including [2, 3, 4], and followed this line to derive the loss function for their DPO, please see Appendix B.2.1 in [4].  Hence, this is a reasonable approach. We will give citations and provide related illustrations.
>
> Moreover, it was pointed out in [5] that "[6] showed that under the Max-Entropy RL formulation, the token-level log-ratio $\log \frac{\pi_{\theta}(y|x)}{\pi_{\text{ref}}(y|x)}$ can be seen as an implicit token-level reward or advantage function (invariant under reward shaping)."
>
> Notably, our derivation is mainly from the relaxation approach in optimization. Although we start from the state-action reward, the final derived loss function coincides with the streamline of DPO.  Specially, when the $\hat{r}(s_t, a_t)$ is not available, we may simply set $\hat{r}(s_t, a_t)=0$ and by Assumption 4.2 our loss function in Equ. (16) is precisely the loss function of DPO, which demonstrates our approach is reasonable.
>
> **Q2 Explanation for $f$ in Assumption 4.2**
>
> Thanks. $f(u)$ is set such that  $f(0)=1$ and  $|f(u) - 1| \le \varepsilon$. Concrete $f(u)$ can be chosen by users for their personal usage. Please also check our response to **Q8 of Reviewer aioX** to see benefits of introducing $f$.
>
> **Q3 Strong assumption for Equ. 15**
>
> Thank you for the insightful comment!
> We have resolved this issue based on the finding that $\delta( f, \hat{r}; x, y_w, y_l)$ does not depend on the policy $\pi_{\theta}$ to be optimized, and the assumptions are removed. The approach is new for canceling partition functions in the BT model.
>
> Now Equ. (15) and its proof have been reorganized as follows:
> $$
>  \Pr(y_w \succ  y_l | x)  = \sigma\left( \varphi(\pi_{\theta}, f, \hat{r}; x, y_w, y_l) + \delta( f, \hat{r}; x, y_w, y_l) \right),  \qquad (1)
> $$
> where  $\delta( f, \hat{r}; x, y_w, y_l)$ does not depend on the policy $\pi_{\theta}$ to be optimized, but only on $f, \hat{r}, x, y_w, y_l$ and the partition function $Z(s_t)$ (does not depend on $\pi_{\theta}$, see Theorem 4.3).
> For briefness, let
> $$h\triangleq (f, \hat{r}; x, y_w, y_l) .$$
>
> Since $\sigma(t)$ is the sigmoid function with $\sigma'(t)>0$, we have:
>
> **Theorem 1.** The $\Pr(y_w \succ  y_l | x)$ in Equ. (1) has the same maximal solution and the same ascent direction as the function $\sigma\left( \varphi(\pi_{\theta}, h)\right)$ with respect to $\pi_{\theta}$.
>
> *Proof.* Note that,  $\delta( h)$ is not dependent on the policy $\pi_{\theta}$ and $\sigma'(t)>0$. $d$ is an ascent direction of function (1) if
> $$ d^T \nabla_{\pi_{\theta}} \sigma\left( \varphi(\pi_{\theta}, h) + \delta(h) \right) >0,$$
> which is equivalent to
> $$
> \begin{aligned}
> &  d^T \sigma'\left( \varphi(\pi_{\theta}, h)+ \delta(h)\right) \nabla_{\pi_{\theta}}  \varphi(\pi_{\theta}, h) >0 \\
> & \Longleftrightarrow  d^T \sigma'\left( \varphi(\pi_{\theta}, h)\right) \nabla_{\pi_{\theta}}  \varphi(\pi_{\theta}, h) >0 \\
> & \Longleftrightarrow d^T \nabla_{\pi_{\theta}}  \sigma\left( \varphi(\pi_{\theta}, h)\right) >0.
> \end{aligned}
> $$
>
> Hence function (1) has the same ascent direction as the function  $\sigma\left( \varphi(\pi_{\theta}, h)\right)$.
> Similarly,
> $$
> \begin{aligned}
> & \nabla_{\pi_{\theta}} \sigma\left( \varphi(\pi_{\theta}, h) + \delta( h) \right) =0  \\
> & \Longleftrightarrow  \sigma'\left( \varphi(\pi_{\theta}, h)\right) \nabla_{\pi_{\theta}}  \varphi(\pi_{\theta}, h) = 0 \\
> & \Longleftrightarrow \nabla_{\pi_{\theta}}  \sigma\left( \varphi(\pi_{\theta}, h)\right)
> = 0.
> \end{aligned}
> $$
> So Theorem holds.
>
> Thus by Thm 1, since we focus only on optimal $\pi_{\theta}$ of function (1), we may redefine
>  $$\Pr(y_w \succ  y_l | x) \triangleq \sigma\left( \varphi(\pi_{\theta}, f, \hat{r}; x, y_w, y_l)\right), $$
> and use it for constructing the loss function in Equ. (16).
>
> We will update the statement and proof of Equ. (15). Thanks!
>
> **Q4 Why $r(s_t,a_t) = \beta \log \pi_{\theta} / \pi_{\text{ref}}$**
>
> It was shown in [6] that $r(s_t,a_t) = \beta \log \pi_{\theta} / \pi_{\text{ref}}$ under the definition of equivalent state-action reward class and invariant re-parameterization, which does not require  $V(s_{t}) = V(s_{t+1})$. Please see Theorem 1 there. Moreover, the final loss function in [6] is equivalent to that in [7]. Hence we adopt the equation directly. It is a common practice in many works [5, 8].
>
> [1] A dense reward view on aligning text-to-image diffusion with preference
>
> [2] Approximately optimal approximate reinforcement learning
>
> [3] Relative entropy policy search
>
> [4] Trust Region Policy Optimization
>
> [5] Self-Play Preference Optimization for Language Model Alignment
>
> [6] From r to Q*: Your language model is secretly a Q-function
>
> [7] Direct preference optimization: Your language model is secretly a reward model
>
> [8] Free process rewards without process labels

---

> > ### Comment · Reviewer_F88J · 2025-04-01
> >
> > Thanks for the rebuttal. Some of my remaining concerns are shown as follows
> >
> > Q1: Thanks for providing the references. However, the question, "the objective in equation (8) is to myopically optimize the reward of the currect step without considering further states but why this objective is applied" is not directly answered.
> >
> > Q2: Thanks the author for the comment. However, it is still unclear to me how should the user choose the function $f$ according to the personal usage. In A8 to reviewer aioX, the author provide an example to show that certain $f$ can emphasize those token with positive reward. But the motivation behind emphasize that token is still unclear.
> >
> > Q3: Thanks the author for the updated proof. However, (potentially due to space limit), the revised proof is hard for me to follow. At a high level, the new proof indicates that $\delta(f, \hat{r}; x, y_w, y_l) =0$, which is quite counter-intuitive.
> >
> > Q4: Thanks for the clarification. My concerns regarding this part are fully addressed.

---

> > > ### Author Response · Authors · 2025-04-06
> > >
> > > **A1:** We must clarify that our final derived equation optimizes the reward of the whole trajectory just like DPO. Equ. (8) is only the starting step for the subsequent derivations.  Indeed, the ground-truth unknown reward  $r_{\phi}(x,y)$ is decomposed into the token level in Equ. (6), and Equ. (8) is for representing the token-level reward $r_{\phi}(s_t, a_t)$ with some policy in the subsequent derivation.
> > >
> > > Specifically, from Equ. (13), the current step $r_{\phi}(s_t, a_t) =  \beta \log\frac {\pi_{\theta} (a_t|s_t)}{\pi_{\text{ref}}(a_t|s_t)} + \beta  \log Z(s_t)$, where the partition function does not depend on $\pi_{\theta}$.  Suppose W.L.O.G. the trajectory generation of LLM is in finite time-steps, then the policies of all time-steps in the equation can be re-parameterized into one policy  $\pi_{\theta^*}$ such that each log-ratio has the same value as the original one, due to huge number of parameters. Then  (for easy presentation, let $f(\cdot)\equiv 1$), it is obvious that:
> > >     $$
> > >     r_{\phi}(x, y) = \sum_{t=1}^T \left[\beta \log\frac {\pi_{\theta^*} (a_t|s_t)}{\pi_{\text{ref}}(a_t|s_t)}+ \beta  \log Z(s_t) \right]
> > >     = \beta \log\frac {\pi_{\theta^*} (y|x)}{\pi_{\text{ref}}(y|x)} + \beta \sum_{t=1}^T  \log Z(s_t).
> > >     $$
> > >
> > > Next, with the Bradley-Terry preference model, the per-instance loss $\sigma( \varphi(\pi_{\theta}) + \delta )$ in Equ. (15) is adopted for maximizing to obtain an optimal policy $\pi_{\theta}$. By Theorem 1, $\sigma( \varphi(\pi_{\theta}) + \delta)$ and $\sigma( \varphi(\pi_{\theta}))$ have the same maxima and ascent directions w.r.t. $\pi_{\theta}$, hence we can redefine
> > >  $$\Pr(y_w \succ  y_l | x) \triangleq \sigma ( \varphi(\pi_{\theta})), $$
> > > and use it to construct the loss function in Equ. (16).  In this case, it is exactly the per-instance loss of DPO.
> > >
> > > **A2:**
> > > **(1) How to choose $f$:** Our proposed $f$ in  Equ. (17) have demonstrated promising performance in RLHF experiments, which is the primary use case for preference optimization algorithms. We believe this makes it a reasonable default choice for similar scenarios. For other use cases, $f$ can be customized accordingly. For example, if smoother guidance is desired, users may use a sigmoid function over $\hat{r}$ in the proposed $f$. Many different choices of $f$ can satisfy Assumption 4.2, and determining the optimal one remains an open problem, promoting further research.
> > >
> > > **(2) Motivation of Equ. (17):** Consider the case in **A8 to reviewer aioX**, if $\hat{r}(s_t, a_t)>0$, i.e., the reward is a positive, then the action $a_t$ in state $s_t$ is preferred. This implies that the state-action $(s_t, a_t)$ should be reinforced, and then it is assigned a larger weight $1 + \alpha \hat{r}([x, y_w^{<t}], y_w^t)$. In this way, the gradient of our loss function $\mathcal{L}\_{\text{TGDPO}}(\pi_{\theta})$ at this state-action is
> > > $$
> > > \beta (1 + \alpha \hat{r}([x, y_w^{<t}], y_w^t))\nabla_{\pi_{\theta}}\log\frac {\pi_{\theta} (y_w^t|[x, y_w^{<t}])}{\pi_{\text{ref}}(y_w^t|[x, y_w^{<t}])},
> > > $$
> > >  which is scaled up by  $1 + \alpha \hat{r}([x, y_w^{<t}], y_w^t)$. As a result, optimizing our loss function encourages the policy to assign higher probability to the action that leads to higher reward in the given state. In contrast, if $\hat{r}(s_t, a_t)<0$, then the action is a dispreferred one, and is progressively assigned lower probability. The other cases are omitted due to limit space. This weight adjustment allows our TGDPO to optimize the policy more effectively, as demonstrated in our experiments.
> > >
> > > **A3:** We must clarify that your high level indication $\delta(f, \hat{r}; x, y_w, y_l) =0$ is not correct.  For example, suppose $\varphi= -t^2$, $\delta=1$, then $\sigma(\varphi )$ and $\sigma(\varphi + \delta)$ have maximizer $t=0$, but in this case $\delta=1$.
> > >
> > > In  **A3 to your Q3 in our previous reply**, we have shown the result in a formal way. For easier understanding, let's give it in a compact form.
> > >
> > > In the main text, Theorem 4.3 shows that the partition function $Z(s_t)$ and $s_t$ do not depend on $\pi_{\theta}$. Moreover, $\delta(\cdot)$ also does not depend on $\pi_{\theta}$. For your understanding, we simplify in the sequel all notations independent of $\pi_{\theta}$ to be optimized, then the Bradley-Terry preference model in Equ. (15) is $\Pr(y_w \succ  y_l | x) = \sigma( \varphi(\pi_{\theta}) + \delta)$, and the Theorem in  **A3 to your Q3 in our previous reply** is exactly as:
> > >
> > > **Theorem 1.** For the policy $\pi_{\theta}$, the function $\sigma( \varphi(\pi_{\theta}) + \delta)$ has the same maxima and ascent directions as the function $\sigma( \varphi(\pi_{\theta}) )$, here $\sigma(t)$ is the sigmoid function.
> > >
> > > Then it is easy to show Theorem 1 is correct since the sigmoid function $\sigma(t)$ is strictly increasing.

---

### Official Review · Reviewer_uVtj · 2025-03-14

**Overall Recommendation:** 3

**Summary:**

This paper introduces TGDPO, an enhanced version of Direct Preference Optimization (DPO) that incorporates token-level reward guidance to address the limitations of conventional sentence-level DPO. While prior methods like Proximal Policy Optimization (PPO) benefit from fine-grained token-level rewards, DPO, formulated as a sentence-level bandit problem, struggles to leverage such granular signals. To bridge this gap, the authors decompose sentence-level PPO into a sequence of token-level PPO problems, enabling the derivation of a closed-form optimal token-level policy and its corresponding token-level reward. By integrating these token-level rewards with the Bradley-Terry preference model, the proposed TGDPO algorithm introduces a new loss function that guides optimization at the token level.
Experiments on MT-Bench, AlpacaEval 2, and Arena-Hard demonstrate TGDPO’s superiority over standard DPO. The method also experimentally exhibits robustness to variations in token-level rewards and provides control over convergence speed.

**Claims And Evidence:**

Please see the Experiment and Question section.

**Essential References Not Discussed:**

The literature review on RLHF could benefit from incorporating additional studies on DPO variants.

**Experimental Designs Or Analyses:**

1. In Tables 1-4, how is the win rate calculated? Why do the scores between the methods appear similar, yet the win rates vary?
2. In Table 2, the number of baselines is relatively small. It would be beneficial to compare against additional baselines, especially TDPO, which is the most closely related work to this paper.
3. Will the choice of $f()$ affect the results? The experiment lack the ablation study of the choice of $f$.
4. In Section 5.3, how is convergence defined? Additionally, how do the authors select checkpoints (no convergence results) in other sections
5. In Tables 2-4, why is the score not reported
6. Are the results stable? It would be better to report the standard deviation.

**Methods And Evaluation Criteria:**

The method is the direct result of the theoretical analysis; please see questions in the Theoretical Claims section.

**Other Comments Or Suggestions:**

No.

**Other Strengths And Weaknesses:**

Strengths:
1. This paper is well-written, and the method is well-motivated by rigorous theoretical analysis. The idea is both novel and interesting.
2. The experiments demonstrate the effectiveness of the method, highlighting additional advantages regarding convergence properties. The authors also provide valuable insights from the experiments.

Weaknesses:
Overall, after carefully reading the paper, I believe this paper meets the acceptance criteria. However, I have several questions regarding the theoretical and experimental sections, which prevent me from confidently voting for acceptance. Please refer to the other sections for detailed questions and concerns.

**Questions For Authors:**

1. In lines 221-223 (left), why is "the token-level reward only used as guidance, and we do not require it to be very accurate"? Wouldn't an inaccurate token-level reward affect the outcome?

2. Why does $\alpha$ affect the convergence rate? Are there any theoretical insights into this?

3. Do the authors have any plans to release the code?

**Relation To Broader Scientific Literature:**

The method might benefit other important aspects of LLM alignment, such as safety, honesty, and fairness.

**Theoretical Claims:**

I have carefully reviewed all the theoretical deductions and proofs, and I have some questions:

1. The modified token-level DPO depends on the choice of $\hat{r}$, which does not seem to be a well-defined problem.

2. In the proof of the Bradley-Terry Model with Token-Level Reward Guidance in Equation 15, the assumption $T_w \approx T_l$ and $Z([x, y_w^{<t}) ~= Z([x, y_l^{<t})$ made by the authors in lines 691-692 is very strong.
However, the data $y_w$ and $y_l$ in the offline dataset used to train the DPO may not be generated from the same model. In practice, we might only have the positive data $y_w$, and we generate $t_l$ through negative sampling, or the pairs
$\{y_w, y_l\}$ could be generated using two different models.

Hence, the assumption that $T_w \approx T_l$ and $Z([x, y_w^{<t}) ~= Z([x, y_l^{<t})$ may not hold in this case.
This could significantly affect the subsequent deductions of the method.

Additionally, the assumption should be mentioned in the main text.

3. In Line 311-313 (left), how can we get this equation? and in practice, should we always need to run DPO first to get the $\hat{r}$.

---

> ### Author Rebuttal · Authors · 2025-04-01
>
> **Q1 TGDPO's dependence on the choice of $\hat{r}$**
>
> Trained token-level rewards have shown effectiveness for PPO [1, 2]. For DPO, it is interesting to ask if there exists a framework that can incorporate a trained token-level reward explicitly for better performance. Our TGDPO fills this nontrivial gap.
>
> **Q2 Strong assumption for Equ. 15**
>
> Thanks for the insightful comment! Now the assumptions have been removed, the approach is novel. The theorem and proof can be seen in Response to **Q3 of Reviewer F88J**. We will make revisions in the final version.
>
> **Q3 How to get the Equation in Lines 311-313 (left)? How to get $\hat{r}$ in practice?**
>
> Thm 1 of [3] shows
> $$r(s_t,a_t) = \beta \log \frac{\pi_{\theta}(a_t|s_t)}{\pi_{\text{ref}} (a_t|s_t)}$$
> under equivalent state-action reward class and invariant re-parameterization. The final loss function in [3] is equivalent to that of DPO [4]. So we adopt the result by setting $s_t= [x, y^{<t}]$ and $a_t= y^t$ and get the Equation. This is a common practice in many works [5].
>
> To obtain a token-level reward $\hat{r}$ in practice, we can use off-the-shelf open-sourced models trained by DPO, other methods [1, 2], or run DPO by ourselves.
>
> Thanks. We will illustrate more for this equation in the final version.
>
> **Q4 Clarification on win rate and score**
>
> The win rate is evaluated by calling a judge model (e.g., gpt-4o-2024-11-20) to pairwise compare the responses of the model and a baseline model (e.g., gpt-4-0314) and determine which one is better [6]. The MT-Bench score is evaluated by calling a judge model to directly assign a score to the model's response [7]. Many prior works observed the scores of different methods being similar [6, 8]. This is likely due to the single-instance scoring protocol.
>
> **Q5 Comparison with TDPO**
>
> Below is the experiment result with TDPO under the Llama3-8B-Instruct PairRM setting.
>
> |  | Arena-Hard win rate |  AlpacaEval 2 win rate | MT-Bench score |  MT-Bench win rate |
> | - | - | - | - | - |
> | SFT | 21.4 |30.6 |7.9 |27.5 |
> | DPO |30.4  | 41.7| **8.0**| 37.5|
> | TDPO | 30.2 | 40.7|**8.0** | 39.0|
> | SimPO |28.7  |39.8 |7.8 | 32.5|
> | TGDPO |  **34.3**| **43.9**| **8.0**|**41.9** |
>
> Result with TDPO using the SFT model OpenRLHF/Llama-3-8b-sft-mixture is in **Q1 of Reviewer GJSH**. The result table with TDPO under other settings is in this link <https://anonymous.4open.science/r/tgdpo_rebuttal/results_with_tdpo.pdf>
>
> **Q6 Will the choice of $f$ affect the results**
>
> $f(u)$ may be adjusted by parameter $\alpha$ as in Equ. (17). With this, experiment results in Fig. 1 reveal in some degree the effects of different choices of $f(u)$. The outcome benchmark difference presented in Fig. 1 and Tab. 3 is subtle.
>
> Our paper proposes a framework for incorporating existing token-level reward into DPO explicitly, with Equ. (17) as an example. Many choices of $f(u)$ can satisfy Assumption 4.2, which one is the best is left as an open problem for stimulating further research.
>
>
> **Q7 Clarification on convergence and checkpoint selection**
>
> In Sec. 5.3, we consider loss moving average below a certain threshold (e.g., 0.1) as convergence. In other sections, we train all methods (DPO, SimPO, TGDPO) for 1 epoch and select the checkpoint at the training end.
>
> **Q8 Stability of results**
>
> Below are the Arena-Hard win rate of Llama3-8B-Instruct PairRM and the 95% confidence interval. From the table we can see all methods have similar levels of stability.
> |  | Arena-Hard win rate |  95% Confidence interval |
> | - | - | - |
> | DPO | 30.4 | (-2.3, 2.2) |
> | SimPO | 28.7 | (-2.0, 2.0) |
> | TDPO | 30.2 | (-2.1, 2.4)|
> | TGDPO | 34.3 |(-2.3, 2.2) |
>
> **Q9 Would error in token-level reward affect results**
>
> By Assumption 4.2, $|1-f(\hat{r}(s_t, a_t))|\le\varepsilon$ where $\varepsilon$ is small. Moreover, by the choice of $f(\hat{r}(s_t, a_t))$ in Equ. (17), the error can be reduced by the parameter $\alpha$. Thus mild errors in token-level reward may not affect the outcome greatly, as demonstrated in Fig. 1 and Tab. 3.
>
> **Q10 Why does $\alpha$ affect the convergence rate**
>
>  Take win response $y_w$ for an example, the other case is similar. In Equ. (16), for $\hat{r}([x, y_w^{<t}], y_w^t)>0$ and larger $\alpha$, the gradient w.r.t. this token is larger, and the convergence would be faster.
>
> **Q11 Code release**
>
> The code will be released upon acceptance.
>
> [1] Preference-grounded token-level guidance for language model fine-tuning
>
> [2] Segmenting text and learning their rewards for improved RLHF in language model
>
> [3] From r to Q*: Your language model is secretly a Q-function
>
> [4] Direct preference optimization: Your language model is secretly a reward model
>
> [5] Free process rewards without process labels
>
> [6] From Crowdsourced Data to High-Quality Benchmarks: Arena-Hard and BenchBuilder Pipeline
>
> [7] Judging LLM-as-a-Judge with MT-Bench and Chatbot Arena
>
> [8] SimPO: Simple Preference Optimization with a Reference-Free Reward

---

### Official Review · Reviewer_GJSH · 2025-03-14

**Overall Recommendation:** 4

**Summary:**

The paper presents TGDPO, a formulation of direct preference optimization (DPO) with an implicit token-level reward instead of an implicit sentence-level reward, and shows that this novel method outperforms standard DPO and provides interpretable training dynamics

More precisely, the paper makes the following contributions and claims:
1) It derives a token-level RL-finetuning objective that is tractable to optimize (Eq10) and apply DPO to derive a token-level DPO loss.
2) TGDPO outperforms DPO and other variants in popular alignment benchmarks.
3) TGDPO provides satisfactory policies when trained until convergence and is robust to the modified token-level reward pre-training phase.

## update after rebuttal
All my concerns have been addressed. It also seems that the other reviewer's concerns have either been addressed or are not critical. I maintain my score from the end of the rebuttal.

**Claims And Evidence:**

Overall, the claims in the paper are backed by enough evidence, but I have concerns about missing information in the experimental setup.

1) The derivation is backed by theoretical results with clear proofs. See the Theoretical claims for details.
2) Evidence is provided in Table 1, but I have concerns regarding the experimental setup. See the Experimental Designs section,
3) This is well presented in Figure 1 and Tables 2, 3, and 4.

**Essential References Not Discussed:**

The authors reference the essential prior work and discuss the differences with the work of Zeng et al. (2024), which is critical to clearly identify the contributions of the paper.

**Experimental Designs Or Analyses:**

It's not clear what the starting point of the TGDPO training is. Is it from the already trained DPO model?
TGDPO seems to use double the budget and DPO as it has to train a policy before starting. The authors do not discuss this computational consideration.

It's not clear why SimPO consistently underperforms DPO in Table 1, although the paper uses a similar experimental setup as Meng et al. (2024). This raises concerns regarding the experimental protocol.

**Methods And Evaluation Criteria:**

I believe that starting from closed-data, fully post-trained models like Llama3-8B-Instruct and Gemma-2-2B-it and then performing further alignment on them gives misleading numbers to the community, as they suggest that the methods presented improve the model further, while what is happening is that they distort their post-training distribution.
The authors do not claim to make the models better than what their original authors did, so this is okay in this paper, but I would strongly recommend the authors to consider only SFT models if possible or add a remark about this.

Otherwise, the choice of evaluation benchmarks and datasets is good. The results would be stronger with an additional dataset, which would be more impactful than varying the reward model, as it only serves for preferences.

**Other Comments Or Suggestions:**

Open to reconsidering my score given input from the authors. My main concern is about missing clarifications for the results and experimental protocol.

**Other Strengths And Weaknesses:**

Clarity:
-The paragraph in lines 215 column two seems grammatically incorrect. It's not clear what point it conveys.
-I would have appreciated more motivation for the Modified Token-Level PPO problem. Its presentation is a bit abrupt.
+Nevertheless, the interpretation of the loss derived from it with the practical considerations makes up for this lack of earlier motivation. Perhaps its possible to rewrite some parts to connect these two motivations?

**Questions For Authors:**

No additional questions.

**Relation To Broader Scientific Literature:**

Sufficient.

**Theoretical Claims:**

I verified the proof of Theorem 4.1. in Appendix A.1.
The theorems to derive the optimal policy and the resulting reward are straightforward modifications of the results in the DPO paper. I did not verify their proofs but believe they hold.
Equation 15 is an important result, but its proof is in the appendix. It would have been nice to provide some intuition in the main paper. I verified the proof. It makes several assumptions, which may or may not hold depending on the data considered. To me, what's important for the paper in this case is that the empirical results still show improvement with this derivation..

---

> ### Author Rebuttal · Authors · 2025-04-01
>
> **Q1 Experiment on SFT model**
>
> Thank you for the advice! Below we show the experiment results on UltraFeedback using the open-sourced SFT model OpenRLHF/Llama-3-8b-sft-mixture, which has not been trained by RLHF. Our TGDPO using SimPO's token-level reward achieves much better performance than baselines. Specifically, it achieves win rate gains of 10.5 on AlpacaEval 2 and 3.9 on Arena-Hard compared to best-performing baselines.
>
> |  | Arena-Hard win rate |  AlpacaEval 2 win rate | MT-Bench score |  MT-Bench win rate |
> | - | - | - | - | - |
> | SFT | 6.2 |5.0 |7.6 |16.3 |
> | DPO | 10.2 |9.9 |**7.8** |19.5 |
> | TDPO | 11.7 |11.0 |7.5 |15.7 |
> | SimPO | 21.4 |16.4 |**7.8** |27.5 |
> | TGDPO w/ DPO's token reward | 13.8 |12.8 |7.7 |20.0 |
> | TGDPO w/ SimPO's token reward | **25.3** |**26.9** |7.6 |**31.9** |
>
> **Q2 Setting difference between SimPO (Meng et al., 2024) and ours**
>
> The key difference in the experiment setting is that we use the latest and more powerful gpt-4o-2024-11-20 as the judge model, while SimPO uses gpt-4-1106-preview (gpt-4 turbo), which was released in Nov 2023.
>
> Below we compare the Arena-Hard win rate of Llama3-8B-Instruct PairRM using these two judge models. The win rate judged by gpt-4-1106-preview is generally consistent with the SimPO paper, while the result is different from the latest gpt-4o-2024-11-20. This is the reason for SimPO underperforming DPO in Table 1.
>
> |  | gpt-4-1106-preview |  gpt-4o-2024-11-20 | Avg |
> | - | - | - | - |
> | DPO | 32.9 |30.4 |31.7 |
> | SimPO | 33.5 |28.7 |31.1 |
> | TGDPO | 36.9 |34.3 |35.6 |
>
> **Q3 Starting point of TGDPO training**
>
> We would like to clarify that DPO, SimPO, and our proposed TGDPO have the same starting point for training, which are Instruct or SFT models. As described in Equation (16), TGDPO is designed to leverage any token-level reward models, including pre-trained DPO or SimPO models. We can use off-the-shelf open-sourced token-level reward models. So it is not necessary to train a token-level reward model by ourselves before starting TGDPO. Furthermore, TGDPO can enjoy faster convergence speed with satisfactory performances, as demonstrated in Figure 1 and Table 3 of this paper. Also, naively increasing the training budget for DPO or SimPO usually does not improve the performance. As demonstrated in Table 2, training DPO and SimPO to convergence leads to worse performance. We will clarify the description to avoid misunderstanding.
>
> **Q4 Clarification of Equ. 15**
>
> We followed the standard way of conducting experiments and TGDPO consistently outperforms baselines. We believe this empirical validation is a key strength of our approach, as it shows that the theoretical insights lead to practical improvements.
>
> From theory aspect, now the related assumptions have been completely removed, the approach is novel, and the intuition is outlined below:
>
> Let
> $$h=( f, \hat{r}; x, y_w, y_l).
> $$
> The Bradley-Terry model in Equ. (15) is
> $$
>  \Pr(y_w \succ  y_l | x)  = \sigma\left( \varphi(\pi_{\theta}, h) + \delta(h) \right).
> $$
> We find that $\delta( h)$ does not depend on the policy $\pi_{\theta}$ to be optimized,  but only on $f, \hat{r}, x, y_w, y_l$ and the partition function $Z(s_t)$ (does not depend on $\pi_{\theta}$, see Theorem 4.3 in the main text). Since $\sigma(t)$ is the sigmoid function with $\sigma'(t)>0$, the above $\Pr(y_w \succ  y_l | x)$ has the same maximal solution and the same ascent direction as the function $\sigma\left( \varphi(\pi_{\theta},h)\right)$ with respect to $\pi_{\theta}$.
>
> Hence, since we focus only on the maximal solution $\pi_{\theta}$, we may redefine
>  $$\Pr(y_w \succ  y_l | x) \triangleq \sigma\left( \varphi(\pi_{\theta}, h)\right), $$
> and use it for constructing the loss function of our preference optimization.
>
> The detail can be seen in response to **Q3 of Reviewer F88J**. We will provide the intuition in the final version.
>
> **Q5 Difference with TDPO (Zeng et al., 2024)**
>
> Our work aims to leverage an existing token-level reward to guide DPO training at the token level. Whereas, TDPO aims to enhance the regulation of KL-divergence by constraining each token with forward KL-divergence. It is not guided by a token-level reward.
>
> We will add more discussions on the differences with TDPO in the final version.
>
> **Q6 Clarification on line 215**
>
> The paragraph demonstrates that it is possible to obtain a token-level reward, using the approach in [1, 2] or DPO. And, the token-level reward will be adopted for shaping the reward in PPO and subsequently the loss function of DPO.
>
> We will make the description better.
>
> [1] Preference-grounded token-level guidance for language model fine-tuning
>
> [2] Segmenting text and learning their rewards for improved RLHF in language model

---

> > ### Comment · Reviewer_GJSH · 2025-04-04
> >
> > I thank the authors for addressing my concerns. One concern remains:
> >
> > > It's not clear why SimPO consistently underperforms DPO in Table 1, although the paper uses a similar experimental setup as Meng et al. (2024).
> >
> > Edit:
> >
> > The authors have cleared the above concern by providing evidence of experiments with the same protocol showing SimPO underperforming DPO. It appears that although these papers all use the same model, dataset, and hyperparameters, they do model selection differently, and often do not report how model selection has been done.
> >
> > I do believe the authors selected the best model for each algorithm with the same criterion, so all my concerns are cleared now. I'm increasing my score from 1 to 4.

---

> > > ### Author Response · Authors · 2025-04-06
> > >
> > > We appreciate the reviewer’s continued engagement and would like to clarify the concern regarding the performance gap between SimPO and DPO for Instruct models using the Ultrafeedback dataset in Table 1 of our main paper.
> > >
> > > As noted previously, the key difference lies in the evaluation setup. While we strictly followed the official implementations and hyperparameter searches of all baselines, we use the most recent and powerful gpt-4o-2024-11-20 as the LLM judge, whereas the original SimPO paper used gpt-4-1106-preview (gpt-4 turbo), released in November 2023. This distinction is critical, as LLM-based evaluation can be sensitive to model versions: different LLM judges may exhibit different preference distributions due to improvements or shifts in alignment objectives.
> > >
> > > To further isolate this factor, we include results in **Q2** using gpt-4-1106-preview, the same judge as in the SimPO paper. Under this setting, SimPO indeed outperforms DPO, consistent with SimPO's observation. However, when evaluated under gpt-4o-2024-11-20, DPO shows better performance. We believe this update in the LLM judge likely contributed to the observed change in relative performance between DPO and SimPO.
> > >
> > > Furthermore, similar behavior has been reported in independent studies for Instruct models using the Ultrafeedback dataset. For instance, in [1], which also uses the UltraFeedback dataset (denoted as Zephyr setting), SimPO is shown to underperform DPO across multiple Instruct models when using the UltraFeedback dataset:
> > >
> > > -   **Table 2 (right column)**: SimPO underperforms DPO on Llama3-8B-Instruct.
> > >
> > > -   **Table 8 (right column)**: SimPO underperforms DPO on Mistral-7B-Instruct.
> > >
> > > -   **Table 7**: SimPO underperforms DPO in multi-iteration preference optimization.
> > >
> > > Importantly, our experiments in **Q1** show that SimPO outperforms DPO for SFT models, which is consistent with [1]. Our TGDPO also achieves better performance using SimPO's token reward in this case.
> > >
> > > Lastly, we emphasize that we treated all baselines fairly and applied consistent settings across all methods. We do not believe the observed performance discrepancy undermines the validity of our experimental protocol or the strength of our contributions since our TGDPO can leverage the token reward from DPO, SimPO, or any other token-level reward models.
> > >
> > > **[1] Paria Rashidinejad and Yuandong Tian. Sail into the Headwind: Alignment via Robust Rewards and Dynamic Labels against Reward Hacking, ICLR 2025.**

---

### Decision · Program_Chairs · 2025-05-01

**Decision:**

Accept (poster)

**Comment:**

The paper introduces TGDPO, which incorporates token-level reward into DPO for preference alignment of LLMs. The main idea is to decompose sentence-level DPO problem into a sequence of token-level problems, enabling the use of more fine-grained reward signals.

Two reviewers recommended acceptance, noting the paper's promising idea and strong empirical results. Two other reviewers had concerns mainly about certain theoretical claims (strong assumption for Equ. 15) and lack of related work discussion (using token reward in DPO is not totally novel).

During rebuttal, the authors added experimental results using an SFT model to alleviate concerns about using post-trained models. They also provided a comparison of different judge models. The mentioned related work is promised to be discussed in the subsequent versions. The only remaining issue seems to be that the author provided a new proof which removes the previous strong assumption for Eq. (15) while this is not fully checked.

Given that one reviewer was ultimately satisfied and the authors provided additional experimental results using an SFT model and a comparison of different judge models in their rebuttal, I recommend an accept.  While concerns about the generalizability of the theoretical framework remain, the empirical results suggest that TGDPO has the potential to improve alignment of large language models.